# SplAttN: Bridging 2D and 3D with Gaussian Soft Splatting and Attention for Point Cloud Completion

Zhaoyang Li [1]   Zhichao You [1]   Tianrui Li [1]

## Abstract

Although multi-modal learning has advanced point cloud completion, the theoretical mechanisms remain unclear. Recent works attribute success to the connection between modalities, yet we identify that standard hard projection severs this connection: projecting a sparse point cloud onto the image plane yields an extremely sparse support, which hinders visual prior propagation, a failure mode we term Cross-Modal Entropy Collapse. To address this practical limitation, we propose SplAttN, which replaces hard projection with Differentiable Gaussian Splatting to produce a dense, continuous image-plane representation. By reformulating projection as continuous density estimation, SplAttN avoids collapsed sparse support, facilitates gradient flow, and improves cross-modal connection learnability. Extensive experiments show that SplAttN achieves state-of-the-art performance on PCN and ShapeNet-55/34. Crucially, we utilize the real-world KITTI benchmark as a stress test for multi-modal reliance. Counter-factual evaluation reveals that while baselines degenerate into unimodal template retrievers insensitive to visual removal, SplAttN maintains a robust dependency on visual cues, validating that our method establishes an effective cross-modal connection. Code is available at https://github.com/zay002/SplAttN.

## 1. Introduction

Point cloud completion is a fundamental challenge in 3D computer vision. While early methods focused on pure geometric reasoning (Yuan et al., 2018; Yang et al., 2018), recent advancements have shifted towards multi-modal strategies (Zhu et al., 2023b; Yu et al., 2024; Lu et al., 2025; Fang et al., 2025) that leverage 2D images as semantic priors. Despite their empirical success, the theoretical underpinning of why and how multi-modality improves completion remains under-explored. Current approaches often proceed without explicit theoretical guidance, utilizing heuristic fusion modules without rigorously defining the statistical advantages of the cross-modal setting.

According to Multimodal Learning Theory (Lu, 2023), the provable advantage of multi-modal learning over uni-modal counterparts hinges on two critical components: Heterogeneity and Connection. Heterogeneity implies that different modalities provide non-redundant information, while Connection refers to the existence of a learnable mapping between modalities. Theoretically, leveraging these properties can improve the generalization bound by a factor of $O(\sqrt{n})$ (Lu, 2023). However, we argue that existing state-of-the-art methods utilizing deterministic hard projection inherently undermine this Connection. By mapping continuous 3D manifolds onto discrete and sparse 2D grids, these methods induce Cross-Modal Entropy Collapse. This sparsity creates a high divergence between the projected features and the true latent distribution required by visual encoders. Consequently, this impedes the gradient flow and limits the ability of the model to learn the optimal connection function between the 2D visual space and the 3D geometric space.

To resolve the potential entropy collapse, we propose SplAttN. Departing from deterministic mappings, it reformulates projection as probabilistic density estimation via Differentiable Gaussian Splatting, replacing the hard projection that collapses a sparse point cloud onto an extremely sparse image-plane support with a dense, continuous representation. Inspired by representation learning across views (Bachman et al., 2019), this formulation ensures that discrete vertices are mapped to a spatially coherent visual density rather than isolated, near-empty pixel locations. Functioning as a differentiable spatial filter, the mechanism effectively bridges the discrete-continuous gap, minimizing alignment errors and enabling the geometric stream to actively query dense visual priors. This query-driven interaction is conceptually related to retrieve-and-compare multimodal reasoning (Yang et al., 2025). Our contributions

---

[1]School of Computing and Artificial Intelligence, Southwest Jiaotong University, Chengdu, China. Correspondence to: Zhichao You <youzc@swjtu.edu.cn>.

*Proceedings of the $43^{rd}$ International Conference on Machine Learning*, Seoul, South Korea. PMLR 306, 2026. Copyright 2026 by the author(s).

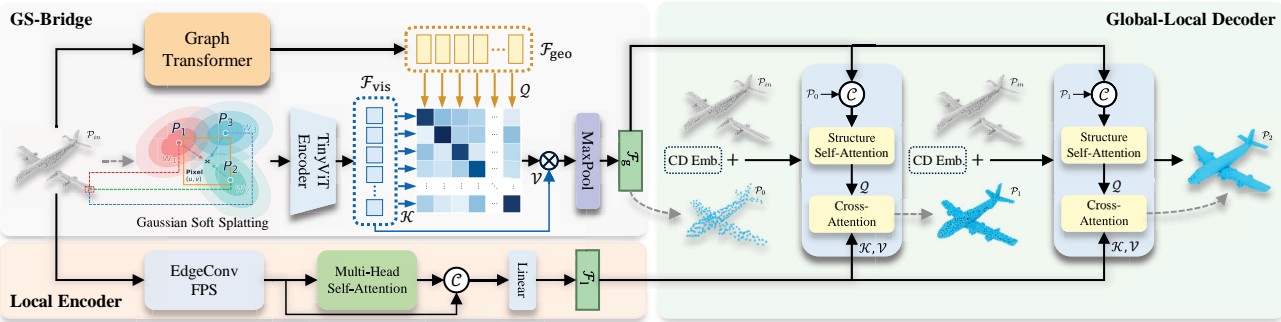

*Figure 1.* **The overall architecture of our proposed SplAttN.** The pipeline consists of two integral stages. **(a) Dual-Branch Feature Extraction.** The GS-Bridge branch extracts comprehensive global representations by using geometric tokens $\mathcal{F}_{geo}$ to actively query visual features $\mathcal{F}_{vis}$ derived from Gaussian Soft Splatting. In parallel, the Local Encoder captures topology-aware local details $\mathcal{F}_l$ through an EdgeConv module followed by Multi-Head Self-Attention and projection. **(b) Global-Local Decoder.** This module unifies the generation process. It first predicts a sparse skeleton $\mathcal{P}_0$ from the global feature $\mathcal{F}_g$ via an MLP, incorporating input priors through the $\mathcal{P}_{in}$-Merge module. Subsequently, it hierarchically upsamples the point cloud ($\mathcal{P}_0 \rightarrow \mathcal{P}_2$). As detailed in the decoding block, each upsampling stage integrates Structure Self-Attention to model geometric consistency and Cross-Attention to inject the extracted local features $\mathcal{F}_l$ (as $\mathcal{K}, \mathcal{V}$) for fine-grained refinement.

extend to a critical verification of multi-modal dependency. While achieving state-of-the-art performance on PCN and ShapeNet-55/34, we leverage the distributional irregularities of KITTI as a stress test for cross-modal reliance. Through a counter-factual evaluation using our Semantic Consistency Score, we reveal that baseline methods effectively degenerate into unimodal template retrievers, showing negligible sensitivity to visual input removal. In contrast, SplAttN demonstrates a strong dependency on visual cues, confirming that our differentiable bridge establishes a bona fide cross-modal connection rather than decoupling generation from observation.

Our main contributions are summarized as follows:

- We ground point cloud completion in Multimodal Learning Theory (Lu, 2023), identifying Cross-Modal Entropy Collapse as the bottleneck restricting the learnable Connection. Crucially, we utilize the KITTI benchmark as a stress test for cross-modal reliance. Through counter-factual evaluation, we empirically verify that SplAttN establishes an effective cross-modal dependency, whereas baseline methods degenerate into unimodal template retrievers.

- We propose SplAttN, a framework that utilizes Differentiable Gaussian Splatting to maximize Point-wise Mutual Information. We theoretically prove that this mechanism functions as a continuous density estimator, strictly expanding valid information support to bridge the modality gap. This reformulation ensures non-vanishing gradients, enabling active and effective alignment between geometric and visual streams.

- We introduce a Hybrid Global-Local Encoder including GS-Bridge and Local Encoder designed to satisfy both local isometry and global homeomorphism. By synergizing graph-based curvature learning with long-range

topological reasoning, it achieves a tighter approximation of the underlying 3D manifold, significantly improving the reconstruction of intricate details and thin structures.

## 2. Related Works

### 2.1. Point Cloud Completion

**Structure-based Methods.** Early Encoder-Decoder works like PCN (Yuan et al., 2018) and FoldingNet (Yang et al., 2018) utilized folding, while TopNet (Tchapmi et al., 2019) used tree decoders. Subsequent methods improved local detail via 3D grids (Xie et al., 2020), iterative refinement (Wang et al., 2020; Yan et al., 2022), and aggregation (Zhang et al., 2020). Others focused on topology via point paths (Wen et al., 2022) or keypoint alignment (Tang et al., 2022).

**Transformer and Generative Architectures.** Transformers reformulated completion as set-to-set translation (Yu et al., 2021; 2023). Variants explore coarse-to-fine generation (Xiang et al., 2021; Zhou et al., 2022), discriminative nodes (Chen et al., 2023; Li et al., 2023), and pure attention (Wang et al., 2024). Recent advances include cross-resolution modeling (Rong et al., 2024), state-space models (Li et al., 2025), and transformers for robust splatting (Chen et al., 2025). However, single-modal methods struggle with semantic ambiguity in severe occlusion.

### 2.2. Cross-Modal and Generative Completion

**Multi-Modal Fusion.** Integrating 2D cues provides semantic priors to resolve geometric ambiguity. Early methods utilized view-guidance (Zhang et al., 2021; Xia et al., 2021), vision-language models (Zhu et al., 2023a), or simple fusion modules (Li et al., 2022; Aiello et al., 2022). Notably,

SVDFormer (Zhu et al., 2023b) and GeoFormer (Yu et al., 2024) project 3D points to query visual features. However, they rely on deterministic hard projection, which induces severe feature sparsity, a phenomenon we identify as Cross-Modal Entropy Collapse. We argue that this sparsity severs the gradient flow, hindering effective utilization of visual information. Consequently, they tend to degenerate into unimodal backbones relying on memorized templates rather than active cross-modal alignment.

**Generative Models.** Diffusion-based approaches (Cheng et al., 2023; Melas-Kyriazi et al., 2023) have achieved remarkable fidelity, with recent innovations even distilling 2D priors from large-scale text-to-image models (Kasten et al., 2023) to guide geometry generation. Nevertheless, their expensive iterative denoising steps incur high latency compared to efficient regression frameworks, limiting their real-time applicability.

### 2.3. Differentiable Rendering and Visual Foundations

**Differentiable Splatting.** Differentiable rendering enables gradient propagation from pixels to geometry, ranging from Softmax Splatting (Niklaus & Liu, 2020) to sphere-based (Lassner & Zollhofer, 2021) and 2D Gaussian surface modeling (Huang et al., 2024). We repurpose 3D Gaussian Splatting (Kerbl et al., 2023) concepts for feature density estimation to bridge the modality gap, effectively transforming discrete point signals into continuous, differentiable feature manifolds.

**Visual Backbones.** Visual encoders evolved from CNNs (He et al., 2016) to Transformers like ViT (Dosovitskiy, 2020) and Swin (Liu et al., 2021). Recent advances like MAE (He et al., 2022) and TinyViT (Wu et al., 2022) further enhance representation efficiency. We address the challenge of utilizing these pre-trained weights on irregular point features via soft splatting, thereby unlocking the potential of transferring large-scale 2D semantic priors to 3D completion tasks.

## 3. Method

### 3.1. Preliminaries

We formulate multi-modal point cloud completion as learning a mapping $\Phi : (\mathcal{P}_{in}, \mathcal{I}) \rightarrow \mathcal{P}_{out}$ to recover the underlying 3D manifold, where $\mathcal{P}_{in} = \{p_i\}_{i=1}^{N} \subset \mathbb{R}^3$ represents the sparse partial observation and $\mathcal{I} \in \mathbb{R}^{H \times W \times 3}$ denotes the dense RGB prior. Let $\mathbf{F}_{geo} \in \mathbb{R}^{N \times C_g}$ and $\mathbf{F}_{vis} \in \mathbb{R}^{H' \times W' \times C_v}$ be the latent geometric tokens and visual feature maps, respectively. Crucially, to rigorously analyze the gradient flow across modalities, we assume a known projection $\pi : \mathbb{R}^3 \rightarrow \Omega \subset \mathbb{R}^2$ and explicitly distinguish three variables: a discrete geometric point $p \in \mathcal{P}_{in}$, its deterministic projected coordinate $\mathbf{u}_p = \pi(p)$, and the

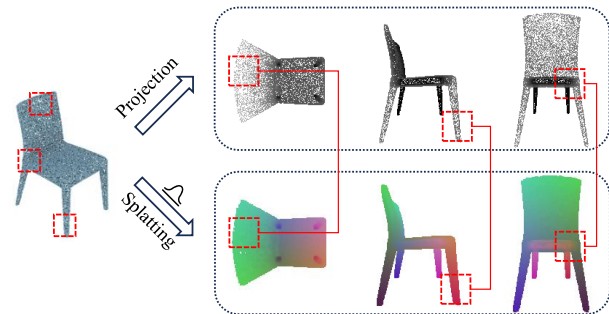

*Figure 2.* **Visualizing the Alignment Gap. Top (Hard Projection):** Hard projection suffers from sparsity and overlap, leading to high divergence from the true manifold. **Bottom (Splatting):** Our method generates a continuous density field, effectively predicting local features for empty regions and smoothing out overlap noise.

continuous spatial query variable $v \in \Omega$ within the visual domain. Standard methods typically model the cross-modal connection by enforcing alignment between $\mathbf{u}_p$ and $v$, but the mathematical formulation of this dependency, whether discrete or continuous, fundamentally determines the differentiability of the system.

### 3.2. Theoretical Analysis

We analyze the limitations of hard projection through its implicit density formulation. Defining the conditional probability of a visual query $v$ given geometry $\mathcal{P}_{in}$ via Dirac delta functions yields:

$$P_{hard}(v|\mathcal{P}_{in}) = \frac{1}{N} \sum_{p \in \mathcal{P}_{in}} \delta(v - \pi(p)) \qquad (1)$$

This formulation fundamentally severs the gradient flow. Considering a loss function $\mathcal{L}$ on the visual domain, the gradient with respect to a geometric point $p$ is derived via the chain rule:

$$\nabla_p \mathcal{L} = \frac{\partial \mathcal{L}}{\partial v} \cdot \frac{\partial v}{\partial \pi(p)} \cdot \nabla_p \delta(v - \pi(p)) \qquad (2)$$

Since the derivative of the Dirac delta is zero almost everywhere, $\nabla_p \mathcal{L} \rightarrow 0$, preventing geometric updates from visual supervision. Furthermore, the support set $\mathcal{S}_{hard} = \{\pi(p)\}$ possesses a Lebesgue measure of zero, $\mu(\mathcal{S}_{hard}) = 0$, leading to entropy collapse.

To resolve this, we reformulate projection as differentiable density estimation using a continuous Gaussian kernel $\mathcal{G}$ with bandwidth $\sigma$:

$$P_{soft}(v|\mathcal{P}_{in}) = \frac{1}{N} \sum_p \alpha_p \mathcal{G}(v; \pi(p), \sigma) \qquad (3)$$

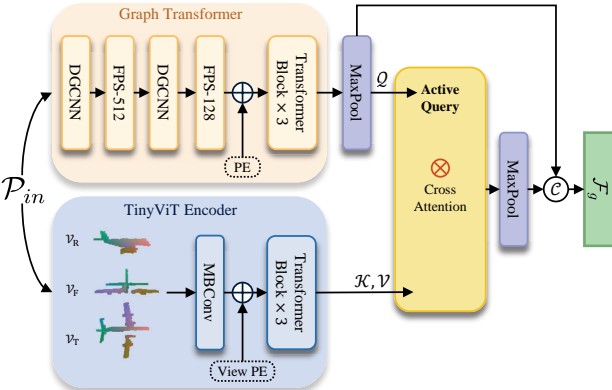

*Figure 3.* **Detailed architecture of the Gaussian Splatting Bridge (GS-Bridge).** It illustrates how the geometric stream interacts with the visual stream through Differentiable Gaussian Splatting to perform density estimation.

This strictly expands the effective information support $\mathcal{S}_{soft} = \bigcup_p \{v \mid \|v - \pi(p)\| < 3\sigma\}$. By the subadditivity of measures, we guarantee positive information capacity:

$$\mu(\mathcal{S}_{soft}) \geq \mu(\mathcal{S}_{hard}) + \sum_{i=1}^{N} (\pi(3\sigma)^2 - \mathcal{O}_{overlap}) > 0 \quad (4)$$

This inequality ensures a non-degenerate probability field with non-vanishing gradients, formally guaranteeing a dense, continuous image-plane support that restores the learnable cross-modal connection and, under an idealized model, implicitly encourages stronger point-wise cross-modal dependency, with a PMI-based interpretation provided in §C.1.

### 3.3. Gaussian Splatting Bridge

We propose the Gaussian Splatting Bridge, a unified differentiable module designed to bridge the discrete-continuous modality gap. It synergizes geometric feature extraction with probabilistic density estimation to establish a learnable connection $g : \mathcal{X} \to \mathcal{Y}$.

#### 3.3.1. HYBRID GEOMETRIC TOKENIZATION

To generate robust geometric queries $\mathbf{F}_{geo}$ capable of actively retrieving visual details, we employ a hybrid architecture that satisfies both local isometry and global homeomorphism.

First, to approximate the complex local surface topology, we extract geometric primitives using EdgeConv. By constructing a dynamic k-Nearest Neighbor graph on the input $\mathcal{P}_{in}$, the EdgeConv operation effectively discretizes the Laplace-Beltrami Operator on the underlying manifold. This allows the network to approximate the local tangent space $T_p\mathcal{M}$

and capture intrinsic mean curvature information:

$$\mathbf{h}_i = \max_{j \in \mathcal{N}(i)} \phi_\theta(p_i, p_j - p_i) \quad (5)$$

where $\phi_\theta$ denotes a shared multi-layer perceptron learning the local surface function, and $\mathcal{N}(i)$ represents the local neighborhood.

While local operators excel at capturing curvature, they struggle with global topological invariants such as holes, symmetry, and disconnected components. To resolve this, we process the local tokens $\mathbf{h}_i$ via a Transformer encoder. The self-attention mechanism functions as a fully connected graphical model, facilitating global message passing to reason about long-range dependencies. The resulting feature set $\mathbf{F}_{geo} \in \mathbb{R}^{N \times C}$ encodes both fine-grained geometric details and global shape semantics.

#### 3.3.2. DIFFERENTIABLE DENSITY IMPLEMENTATION

Guided by the theoretical density formulation in Eq. 3, we implement the continuous visual manifold reconstruction via Differentiable Gaussian Soft Splatting. This process transforms the discrete visual feature map $\mathbf{F}_{img}$ into a continuous density field.

For an arbitrary spatial query $\mathbf{q}$, representing a sub-pixel location on the visual plane, we define the aggregated feature $\mathcal{V}(\mathbf{q})$ as the normalized weighted expectation of the projected primitives:

$$\mathcal{V}(\mathbf{q}) = \frac{\sum_{k \in \mathcal{N}(\mathbf{q})} w_k(\mathbf{q}) \cdot f_k}{\sum_{k \in \mathcal{N}(\mathbf{q})} w_k(\mathbf{q}) + \epsilon} \quad (6)$$

where $\mathcal{N}(\mathbf{q})$ denotes the set of projected primitives contributing to the query location $\mathbf{q}$, $w_k(\mathbf{q})$ is the soft aggregation weight assigned to the $k$-th primitive, and $f_k$ is the feature attached to that primitive. In our CCM implementation, $f_k$ is concretely instantiated as a three-channel pseudo-color derived from normalized 3D coordinates.

The weight $w_k(\mathbf{q})$ is carefully designed to address two fundamental challenges in 2D-3D projection, namely misalignment noise and occlusion. It is formulated as the product of a spatial kernel and a depth prior:

$$w_k(\mathbf{q}) = \underbrace{\exp\left(-\frac{\|\mathbf{u}_k - \mathbf{q}\|^2}{2\sigma^2}\right)}_{\mathcal{G}:\text{Spatial Low-Pass Filter}} \cdot \underbrace{(z_k + \epsilon)^{-1}}_{\mathcal{D}:\text{Soft Z-Buffer}} \quad (7)$$

The Gaussian kernel $\mathcal{G}$ acts as a spatial smoother. It suppresses high-frequency noise caused by quantization errors during projection and, more importantly, provides a smooth gradient landscape. Unlike Dirac delta functions, the Gaussian tail ensures that gradients $\nabla_{\mathbf{u}}\mathcal{L}$ are non-vanishing even when points are slightly misaligned, enabling effective back-propagation to update geometric coordinates. The inverse

depth term $\mathcal{D}$ assigns higher importance to points closer to the camera, corresponding to smaller $z_k$. This effectively approximates a continuous, differentiable Z-buffer, allowing the network to prioritize foreground geometry while maintaining differentiability, which is lost in standard hard z-buffering.

### 3.3.3. ACTIVE CROSS-MODAL ALIGNMENT

With the densified visual field $\mathcal{V}$, we employ Active Attention to functionally implement this PMI objective and establish the cross-modal connection. In contrast to passive concatenation, we treat extracted geometric features $\mathbf{F}_{geo}$ as Queries, and the visual manifold $\mathcal{V}$ as Keys and Values. The network dynamically retrieves relevant visual context:

$$\mathbf{F}_g = \mathbf{F}_{geo} + \text{Softmax}\left(\frac{(\mathbf{F}_{geo}\mathbf{W}_Q)(\mathcal{V}\mathbf{W}_K)^T}{\sqrt{d}}\right)(\mathcal{V}\mathbf{W}_V)$$
(8)

This formulation functions as a differentiable dictionary lookup. By calculating the similarity matrix between geometric structure and visual patterns, the model explicitly learns where to look in the image to refine specific 3D parts. This active querying capability allows the geometry to selectively assimilate semantic priors, mitigating the impact of background clutter and maximizing the flow of valid mutual information.

### 3.4. Global-Local Decoder

We design a Global-Local Decoder to hierarchically densify the coarse skeleton $\mathcal{P}_0$ into $\mathcal{P}_1$ and $\mathcal{P}_2$. As shown in Figure 4, this module integrates structural priors with the local context $\mathbf{F}_l$ through a dual-branch mechanism.

**Uncertainty-Aware Feature Query.** Following SVD-Former (Zhu et al., 2023b), we employ a Structure Analysis unit. We interpret the Chamfer Distance between upsampled points and the input as a proxy for local reconstruction uncertainty. Projecting this geometric error into high-dimensional embeddings enables the self-attention block to spatially modulate feature density, explicitly highlighting regions with high geometric entropy, namely, missing parts.

**Active Local Refinement.** To recover fine details, we utilize a Similarity Alignment module via Multi-Head Cross-Attention. Here, structure-enhanced features act as the Query to retrieve geometric context from the hybrid local primitives $\mathbf{F}_l$ (Key/Value) This operation functions as a differentiable dictionary lookup, anchoring the refinement in high-frequency curvature information captured by the EdgeConv branch.

**Residual Manifold Learning.** We concatenate the outputs from both branches to fuse global structural guidance with local texture. This fused representation is processed by a convolution-based decoding head to expand

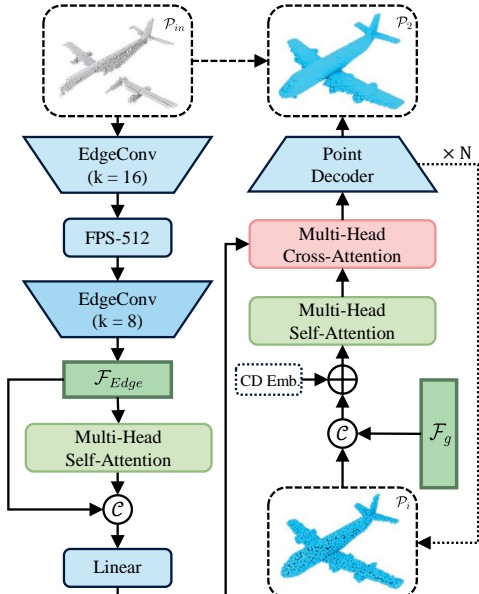

*Figure 4.* **Architecture of the Global-Local Decoder.** The decoder combines global priors with local details. It employs structure-aware attention to query local geometric primitives from the Hybrid Tokenizer for coordinate refinement.

feature resolution and regress a continuous *displacement field* $\psi : \mathcal{P}_k \rightarrow \mathcal{P}_{k+1}$. The predicted coordinate offsets $\Delta\mathcal{P}$ project the coarse approximation onto the high-fidelity manifold via residual learning.

### 3.5. Loss Function

We implement the Chamfer Distance (CD) as the fundamental reconstruction objective. Given two point sets $X$ and $Y$:

$$\mathcal{L}_{\text{CD}}(X,Y) = \frac{1}{|X|}\sum_{x\in X}\min_{y\in Y}\|x-y\|_2^2 + \frac{1}{|Y|}\sum_{y\in Y}\min_{x\in X}\|x-y\|_2^2$$
(9)

To address outlier sensitivity (Lin et al., 2023) and balance loss magnitudes in hierarchical generation, we employ the Weighted Arc-CD ($\mathcal{L}_{\text{warc}}$) via a hyperbolic transformation:

$$\mathcal{L}_{\text{warc}}(X,Y;\lambda) = \lambda \cdot \text{arccosh}\left(1 + \mathcal{L}_{\text{CD}}(X,Y)\right)$$
(10)

The non-linearity naturally compresses outliers while maintaining fine-grained sensitivity. By setting uniform scalar weights $\lambda_k = 1.0$ across all stages $\mathcal{P}_{0,1,2}$, the total training objective is defined as:

$$\mathcal{L}_{total} = \mathcal{L}_{\text{warc}}(\mathcal{P}_0, \mathbf{P}_{gt}; \lambda_0) + \sum_{k=1}^{2}\mathcal{L}_{\text{warc}}(\mathcal{P}_k, \mathbf{P}_{gt}; \lambda_k)$$
(11)

*Table 1.* **Quantitative comparison on the PCN dataset.** We report $L_1$ Chamfer Distance (CD), Density-aware Chamfer Distance (DCD), and F1-Score (F1). CD and DCD are scaled by $10^3$. The best results are highlighted in **bold**.

| Method | CD-Avg ↓ | DCD-Avg ↓ | F1 ↑ | Plane | Cabinet | Car | Chair | Lamp | Sofa | Table | Boat |
|---|---|---|---|---|---|---|---|---|---|---|---|
| FoldingNet (Yang et al., 2018) | 14.31 | - | - | 9.49 | 15.80 | 12.61 | 15.55 | 16.41 | 15.97 | 13.65 | 14.99 |
| TopNet (Tchapmi et al., 2019) | 12.15 | - | - | 7.61 | 13.31 | 10.90 | 13.82 | 14.44 | 14.78 | 11.22 | 11.12 |
| PCN (Yuan et al., 2018) | 9.64 | - | 0.695 | 5.50 | 22.70 | 10.63 | 8.70 | 11.00 | 11.34 | 11.68 | 8.59 |
| GRNet (Xie et al., 2020) | 8.83 | 0.622 | 0.708 | 6.45 | 10.37 | 9.45 | 9.41 | 7.96 | 10.51 | 8.44 | 8.04 |
| SnowflakeNet (Xiang et al., 2021) | 7.21 | 0.585 | 0.801 | 4.29 | 9.16 | 8.08 | 7.89 | 6.07 | 9.23 | 6.55 | 6.40 |
| 3DMamba (Li et al., 2025) | 6.91 | - | 0.805 | 3.86 | 9.12 | 7.72 | 7.41 | 5.73 | 9.04 | 6.29 | 6.09 |
| PointAttN (Wang et al., 2024) | 6.84 | - | - | 3.88 | 9.01 | 7.60 | 7.28 | 5.97 | - | - | - |
| SeedFormer (Zhou et al., 2022) | 6.74 | 0.583 | 0.818 | 3.85 | 9.05 | 7.90 | 7.38 | 5.82 | 8.85 | 6.35 | 6.18 |
| AnchorFormer (Chen et al., 2023) | 6.59 | - | - | 3.70 | 8.94 | 7.57 | 7.05 | 5.21 | 8.40 | 6.03 | 5.81 |
| SVDFormer (Zhu et al., 2023b) | 6.54 | 0.536 | 0.841 | 3.62 | 8.79 | 7.46 | 6.91 | 5.33 | 8.49 | 5.90 | 5.83 |
| AdaPoinTr (Yu et al., 2023) | 6.53 | - | 0.845 | 3.68 | 8.82 | 7.47 | 6.85 | 5.47 | 8.35 | 5.80 | 5.76 |
| GeoFormer (Yu et al., 2024) | 6.42 | 0.526 | 0.853 | 3.61 | 8.70 | 7.46 | 6.72 | 5.14 | 8.27 | 5.84 | 5.64 |
| **Ours** | **6.36** | **0.523** | **0.854** | **3.54** | **8.69** | **7.45** | **6.54** | **5.11** | **8.25** | **5.78** | **5.57** |

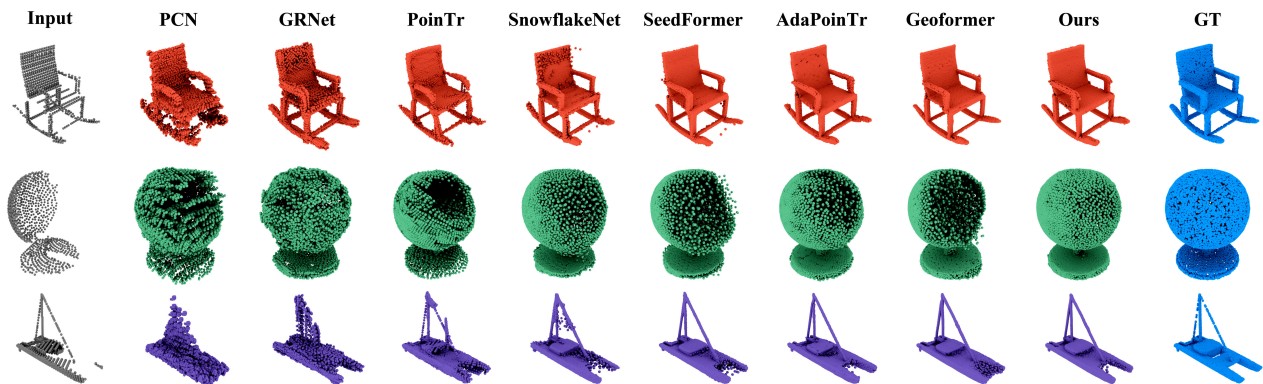

*Figure 5.* **Visual comparison on the PCN dataset.** Compared with state-of-the-art methods, SplAttN recovers more faithful global topology and finer local details, particularly in thin structures like chair legs, verifying the effectiveness of our Hybrid Local Encoder.

## 4. Experiment

### 4.1. Datasets and Metrics

We evaluate SplAttN on three standard benchmarks: PCN, ShapeNet-55/34, and KITTI.

**PCN Dataset (Yuan et al., 2018).** Derived from 8 categories, it contains 30,974 point cloud pairs generated via back-projecting depth images to simulate occlusion. We follow the standard split with 29,671 training, 103 validation, and 1,200 testing samples.

**ShapeNet-55/34 Dataset (Yu et al., 2021).** This benchmark covers a broader taxonomy with 55 categories. It includes 41,952 training samples (ShapeNet-55) and 10,518 testing samples (ShapeNet-34). The test set is stratified into Simple, Medium, and Hard levels based on missing ratios.

**KITTI Dataset (Geiger et al., 2013).** We utilize the KITTI dataset to empirically verify our theoretical propositions regarding cross modal dependency. By applying the model trained on PCN directly to 2401 real world car instances without fine tuning we probe whether the network maintains valid multi-modal connections or degenerates into unimodal

template retrieval when facing distinct distribution shifts.

**Implementation and Metrics.** Our method is implemented in PyTorch and trained on four NVIDIA RTX 4090 GPUs. Intuitively, we set the kernel size of the Gaussian Splatting to 4. We optimize the network using the AdamW optimizer (Loshchilov & Hutter, 2017), where the learning rate is dynamically adjusted via a one-cycle cosine annealing strategy (Smith, 2017) to ensure stable convergence. For evaluation, we employ CD as the primary metric. Following standard conventions (Yuan et al., 2018; Yu et al., 2024), we report the $L_1$-CD scaled by $10^3$ for the PCN dataset. For ShapeNet-55/34, we report the $L_2$-CD scaled by $10^3$ and F-Score@1% to measure reconstruction fidelity. In all comparative tables, methods are listed in descending order of average CD.

### 4.2. Comparison with State-of-the-Art Methods

**Performance on PCN Dataset.** As shown in Table 1, SplAttN achieves state-of-the-art performance with an average CD of 6.36. Unlike methods relying on restrictive symmetry priors, our unified architecture demonstrates supe-

*Table 2.* **Quantitative comparison on the ShapeNet-55 dataset.** We report $L_2$ Chamfer Distance (CD) scaled by $10^3$ and F-Score@1% (F1). **CD-S**, **CD-M**, and **CD-H** denote the CD scores under Simple, Medium, and Hard difficulty levels, respectively. The leftmost ten columns report the CD performance on representative categories. The best results are highlighted in **bold**.

| Method | Table | Chair | Plane | Car | Sofa | Bird | Bag | Remote | Key | Rocket | CD-S | CD-M | CD-H | **CD-Avg** ↓ | F1 ↑ |
|---|---|---|---|---|---|---|---|---|---|---|---|---|---|---|---|
| FoldingNet (Yang et al., 2018) | 2.53 | 2.81 | 1.43 | 1.98 | 2.48 | 4.71 | 2.79 | 1.44 | 1.24 | 1.48 | 2.67 | 2.66 | 4.05 | 3.12 | 0.082 |
| TopNet (Tchapmi et al., 2019) | 2.21 | 2.53 | 1.14 | 2.18 | 2.36 | 4.83 | 2.93 | 1.49 | 0.95 | 1.32 | 2.26 | 2.16 | 4.30 | 2.91 | 0.126 |
| PCN (Yuan et al., 2018) | 2.13 | 2.29 | 1.02 | 1.85 | 2.06 | 4.50 | 2.86 | 1.33 | 0.89 | 1.32 | 1.94 | 1.96 | 4.08 | 2.66 | 0.133 |
| GRNet (Xie et al., 2020) | 1.63 | 1.88 | 1.02 | 1.64 | 1.72 | 2.97 | 2.06 | 1.09 | 0.89 | 1.03 | 1.35 | 1.71 | 2.85 | 1.97 | 0.238 |
| SnowflakeNet (Xiang et al., 2021) | 1.05 | 1.27 | 0.68 | 1.18 | 1.05 | 2.16 | 1.33 | 0.71 | 0.49 | 0.72 | 0.82 | 1.18 | 2.21 | 1.41 | 0.343 |
| PoinTr (Yu et al., 2021) | 0.81 | 0.95 | 0.44 | 0.91 | 0.79 | 1.86 | 0.93 | 0.53 | 0.38 | 0.57 | 0.58 | 0.88 | 1.79 | 1.09 | 0.464 |
| CRA-PCN (Rong et al., 2024) | 0.66 | 0.74 | 0.37 | 0.85 | 0.66 | 1.36 | 0.73 | 0.43 | 0.35 | 0.50 | 0.48 | 0.71 | 1.37 | 0.85 | - |
| SVDFormer (Zhu et al., 2023b) | 0.63 | 0.72 | 0.38 | 0.79 | 0.64 | 1.36 | 0.74 | 0.44 | 0.32 | 0.44 | 0.48 | 0.71 | 1.28 | 0.82 | 0.444 |
| **Ours** | **0.57** | **0.65** | **0.33** | **0.69** | **0.55** | **1.29** | **0.60** | **0.38** | **0.29** | **0.44** | **0.41** | **0.64** | **1.27** | **0.77** | **0.520** |

*Table 3.* **Generalization performance on ShapeNet-34/21.** We report $L_2$ Chamfer Distance (CD, scaled by $10^3$) and F-Score@1% (F1) on 34 seen categories and 21 unseen categories. **CD-S/M/H** denote Simple, Medium, and Hard splits. SplAttN demonstrates superior generalization on unseen classes.

| Methods | 34 Seen Categories | | | | | 21 Unseen Categories | | | | |
|---|---|---|---|---|---|---|---|---|---|---|
| | CD-S | CD-M | CD-H | CD-Avg ↓ | F1 ↑ | CD-S | CD-M | CD-H | CD-Avg ↓ | F1 ↑ |
| FoldingNet (Yang et al., 2018) | 1.86 | 1.81 | 3.38 | 2.35 | 0.139 | 2.76 | 2.74 | 5.36 | 3.62 | 0.095 |
| PCN (Yuan et al., 2018) | 1.87 | 1.81 | 2.97 | 2.22 | 0.150 | 3.17 | 3.08 | 5.29 | 3.85 | 0.101 |
| TopNet (Tchapmi et al., 2019) | 1.77 | 1.61 | 3.54 | 2.31 | 0.171 | 2.62 | 2.43 | 5.44 | 3.50 | 0.121 |
| GRNet (Xie et al., 2020) | 1.26 | 1.39 | 2.57 | 1.74 | 0.251 | 1.85 | 2.25 | 4.87 | 2.99 | 0.216 |
| PoinTr (Yu et al., 2021) | 0.76 | 1.05 | 1.88 | 1.23 | 0.421 | 1.04 | 1.67 | 3.44 | 2.05 | 0.384 |
| SeedFormer (Zhou et al., 2022) | 0.48 | 0.70 | 0.30 | 0.83 | 0.452 | 0.61 | 1.07 | 2.35 | 1.34 | 0.402 |
| PointAttN (Wang et al., 2024) | 0.51 | 0.70 | 1.23 | 0.81 | - | 0.76 | 1.15 | 2.23 | 1.38 | - |
| SVDFormer (Zhu et al., 2023b) | 0.46 | 0.65 | 1.13 | 0.75 | 0.457 | 0.61 | 1.05 | 2.19 | 1.28 | 0.427 |
| AdaPoinTr (Yu et al., 2023) | 0.48 | 0.63 | 1.07 | 0.73 | 0.469 | 0.61 | 0.96 | **2.11** | 1.23 | 0.416 |
| **Ours** | **0.38** | **0.56** | **1.05** | **0.65** | **0.533** | **0.53** | **0.96** | 2.19 | **1.22** | **0.481** |

rior flexibility, particularly in complex categories like *Chair* (6.54 vs. 6.71 of GeoFormer). This verifies that our Hybrid Local Encoder effectively resolves intricate topological structures through intrinsic feature learning.

**Performance on ShapeNet55/34.** Table 2 reports ShapeNet-55 results using **mean class** aggregation. SplAttN achieves the highest F1-Score of 0.520 and surpasses SVD-Former with an average CD of 0.77. Crucially, our method dominates on data-rich head classes (e.g., 0.33 CD on *Plane*) while demonstrating superior robustness on tail categories, significantly outperforming SVDFormer on *Birdhouse* (1.29 vs. 1.36) and *Bag* (0.60 vs. 0.74) as visualized in Figure 6.

Table 3 extends evaluation to ShapeNet-34/21. SplAttN secures the best F1-Score (0.533) and lowest average CD on both seen (0.65) and unseen (1.22) splits, consistently outperforming competitors like AdaPoinTr (1.23) and SVD-Former (1.28). This global superiority, consistent with the entropy gains quantified in Figure 8, validates that maximizing information throughput directly improves geometric reconstruction, with additional qualitative visualizations presented in §A.

**Rethinking the KITTI Benchmark.** Recent studies (Yan et al., 2025) indicate that standard metrics like Fidelity Distance (FD) and MMD correlate poorly with perceptual quality, often favoring generic shape retrieval over faithful and

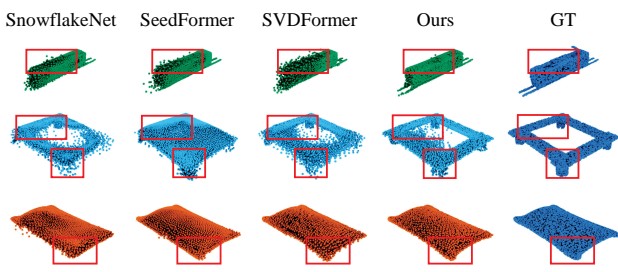

*Figure 6.* **Qualitative comparison on ShapeNet-55.** SplAttN generates more complete and detailed shapes compared to the former baselines across diverse categories.

structurally precise reconstruction. Rather than viewing KITTI merely as a target for domain adaptation, we identify a unique opportunity within its distributional irregularities and intrinsic data imperfections. We argue that the intrinsic artifacts of real-world LiDAR, specifically its extreme sparsity and ray-like anisotropy as visualized in Figure 7, provide an ideal stress test environment for evaluating feature robustness. This distinctive distribution, which starkly contrasts with the uniform sampling of synthetic training data, allows us to rigorously verify whether a multi-modal model truly generalizes via cross-modal connection or simply degenerates into retrieving memorized 3D templates.

To disentangle the actual contribution of visual priors from

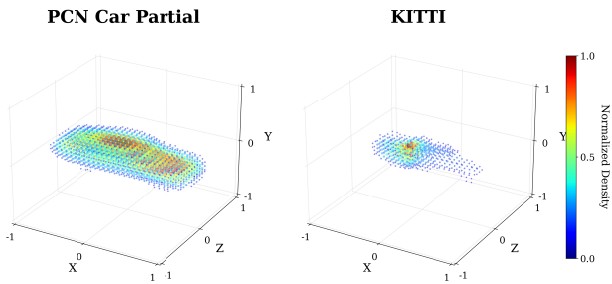

*(a)* **3D Point Density.** Note the ray-like sparsity in KITTI.

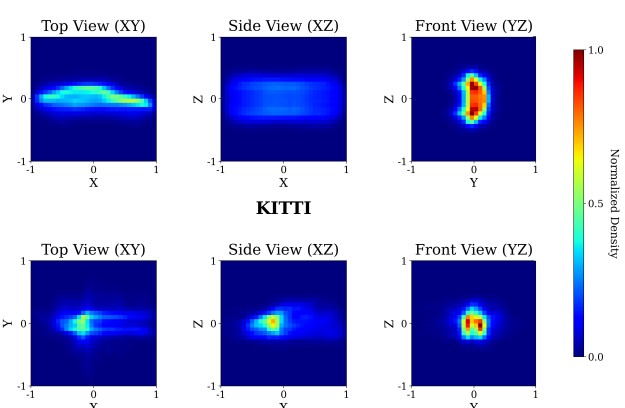

*(b)* **2D Projection Profile.** KITTI exhibits severe signal fragmentation.

*Figure 7.* **Distributional Discrepancy.** Visual comparison of (a) 3D density and (b) 2D projections between PCN and KITTI. The stark contrast reveals a fundamental topological gap, challenging the validity of standard normalization-based evaluation protocols.

geometric memorization, we design a systematic counterfactual evaluation protocol. We employ the Semantic Consistency Score (SCS) as a measure of recognizability, defined by the confidence of a pre-trained oracle classifier on the reconstructed output.

Crucially, we introduce a baseline metric, SCS*, computed by explicitly severing the visual connection, such as zeroing out the input to the 2D branch to effectively isolate geometric signals. This comparison reveals the true dependency of the model on visual signals during the inference process.

As illustrated in Figure 8, severing the visual branch in SVDFormer results in a negligible performance fluctuation ($+0.4\%$), demonstrating a distinct lack of sensitivity. This implies that the model has effectively degenerated into a uni-modal 3D backbone, hallucinating shapes based on learned dataset priors rather than the input observation.

Conversely, GeoFormer exhibits an anomalous performance gain ($+20.9\%$) without images, suggesting that its hard projection mechanism fails to process the domain-shifted visual data, treating it as noise interference. In stark contrast, SplAttN experiences a precipitous drop in consistency

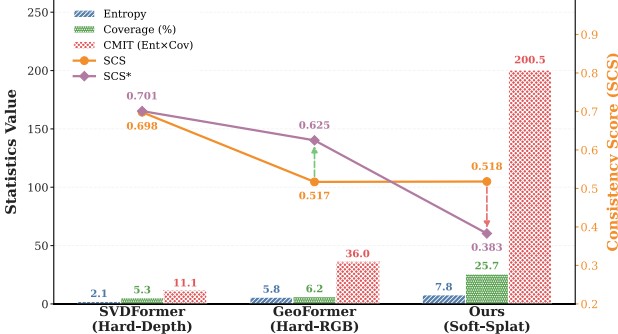

*Figure 8.* **Verification of Multi-Modal Dependency.** We compare SCS sensitivity against Cross-Modal Information Throughput (CMIT). Unlike baselines with low CMIT showing negligible sensitivity, SplAttN achieves a dominant CMIT of **200.5**. This high throughput strictly correlates with a substantial consistency drop upon visual removal, confirming a valid cross-modal dependency rather than template retrieval.

($-26.1\%$) when visual cues are removed. This significant decay empirically validates a genuine cross-modal dependency, which is theoretically underpinned by the high average Cross-Modal Information Throughput (CMIT) of **200.5** (see §C.3) on KITTI, confirming that our bridge actively maximizes information flow to guide reconstruction.

*Table 4.* **Effect of Projection & Geometry.** Comparison on PCN.

| Proj. Strategy | Arch. | | Metrics | | |
|---|---|---|---|---|---|
| | Conv | Hyb. | CD↓ | DCD↓ | F1↑ |
| Hard (Depth) | ✓ | | 6.59 | 0.535 | 0.846 |
| Hard (CCM) | ✓ | | 6.56 | 0.537 | 0.845 |
| Splatting | ✓ | | 6.48 | 0.528 | 0.848 |
| Hard (Depth) | | ✓ | 6.43 | 0.527 | 0.853 |
| Hard (CCM) | | ✓ | 6.41 | 0.529 | 0.853 |
| **Splatting (Ours)** | | ✓ | **6.36** | **0.523** | **0.854** |

*Table 5.* **Visual Encoder Analysis.** Impact of model scale and pre-training on PCN.

| Visual Encoder Settings | | Metrics | |
|---|---|---|---|
| Backbone | Pretrained Weight | CD↓ | F1↑ |
| ResNet-18 | None | 6.44 | 0.849 |
| TinyViT-5M | None | 6.39 | 0.850 |
| TinyViT-5M | IN-1k | 6.37 | 0.852 |
| TinyViT-5M | IN-22k | 6.37 | 0.853 |
| **TinyViT-5M** | **IN-22k→1k** | **6.36** | **0.854** |
| TinyViT-11M | None | 6.39 | 0.849 |
| TinyViT-11M | IN-22k→1k | 6.37 | 0.852 |
| TinyViT-21M | IN-22k→1k | 6.42 | 0.852 |

### 4.3. Ablation Study

We validate our design choices on the PCN dataset.

**Projection Strategy and Geometric Backbone.** Table 4 shows Differentiable Splatting outperforms hard projections (CD 6.36) by modeling soft distributions. CCM yields gains over Depth (6.41 vs 6.43), confirming explicit 3D coordinates reduce ambiguity effectively. Furthermore, the Hybrid architecture surpasses the Convolutional baseline, validating the need for global attention.

**Visual Encoder Analysis.** Table 5 indicates that pre-trained TinyViT-5M significantly improves reconstruction surpassing the ResNet-18 baseline (CD 6.36). Conversely, scaling to 21M degrades performance to 6.42. This implies over-parameterization on the PCN dataset, leading to overfitting on high-frequency noise. Thus, we adopt the 5M model for its optimal balance.

**Computational Cost.** We provide a detailed comparison of computational cost, including parameter count, MACs, inference latency, and GPU memory usage, in Appendix E.

## 5. Conclusion

We demonstrate that resolving Cross-Modal Entropy Collapse, by expanding the support set of 2D image information via differentiable density estimation, is fundamental to bridging the gradient gap in sparse geometric completion. Beyond achieving state-of-the-art performance on PCN and ShapeNet-55/34, our counter-factual analysis on KITTI verifies that SplAttN establishes a robust cross-modal connection, unlike baselines that degenerate into unimodal template retrieval. Future work will target unsupervised domain adaptation and backbone lightweighting to improve scalability. Furthermore, we aim to investigate advanced fusion mechanisms that explicitly enhance inter-modal alignment while mitigating information redundancy, ensuring more compact and efficient multi-modal representations.

## Acknowledgement

This work was supported by the Sichuan Science and Technology Program (Grant Nos. 2025ZDZX0027, 2024YFCY0021, 2025ZNSFSC1279, 2024NSFTD0036), the National Natural Science Foundation of China (Grant No. 5257056442), and the Fundamental Research Funds for the Central Universities (Grant No. 2682026ZT007, 2682025CX012), and Fundamental and Interdisciplinary Disciplines Breakthrough Plan of the Ministry of Education of China (JYB2025XDXM211).

## Impact Statement

This paper presents work whose goal is to advance the field of Machine Learning, specifically in 3D point cloud completion and multi-modal learning. Potential societal consequences of our work include improvements in autonomous driving perception systems and robotic manipulation, which could enhance safety and efficiency. We do not feel there are specific ethical issues that must be highlighted here.

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

# A. Additional Qualitative Results on ShapeNet-55

We provide comprehensive qualitative visualization results on the ShapeNet-55 dataset across varying difficulty levels. As illustrated in Figure 9 (Easy), Figure 10 (Median), and Figure 11 (Hard), baseline methods like SVDFormer often produce over-smoothed shapes or lose local details. In contrast, SplAttN successfully preserves fine-grained geometric structures and accurately recovers missing regions, demonstrating robust performance ranging from simple structures to challenging scenarios with severe occlusion.

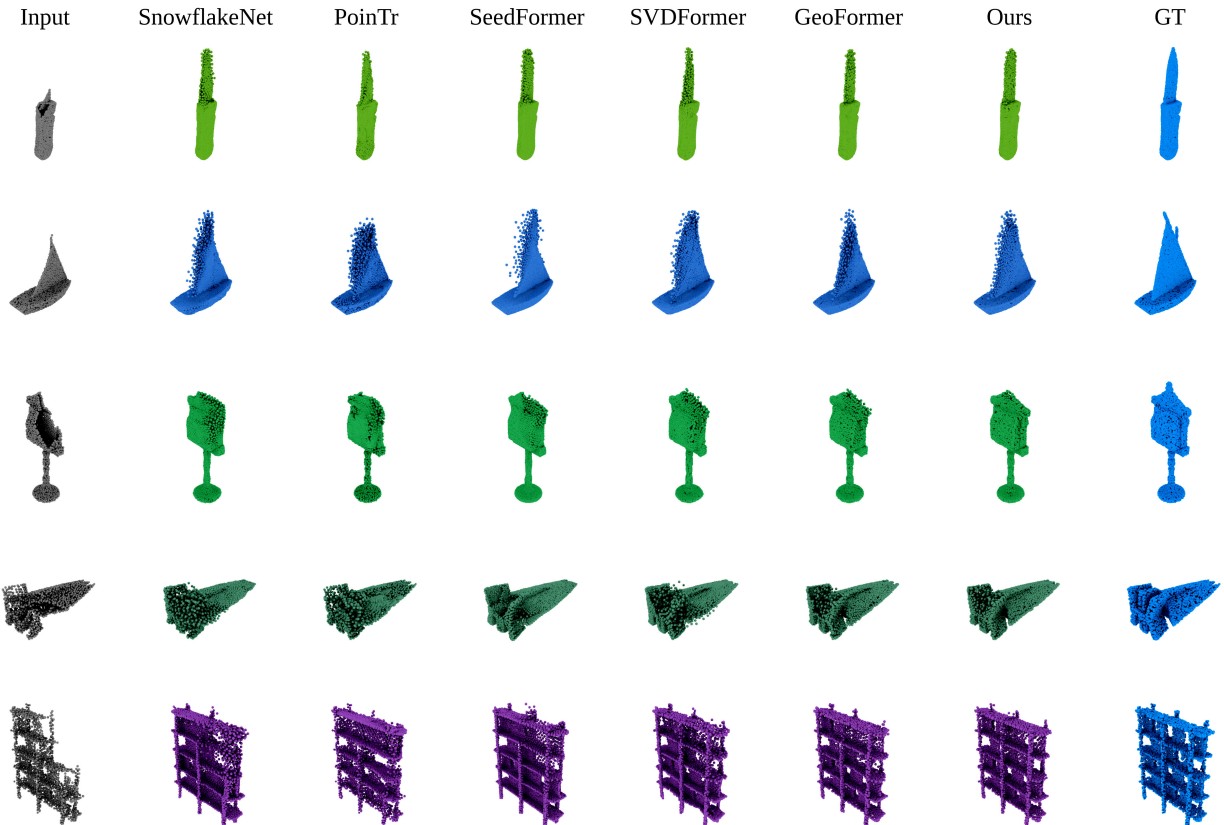

*Figure 9.* **Qualitative Results on ShapeNet-55 (Easy Difficulty).** Comparisons of reconstruction quality on representative samples. SplAttN faithfully recovers details that are blurred by baselines.

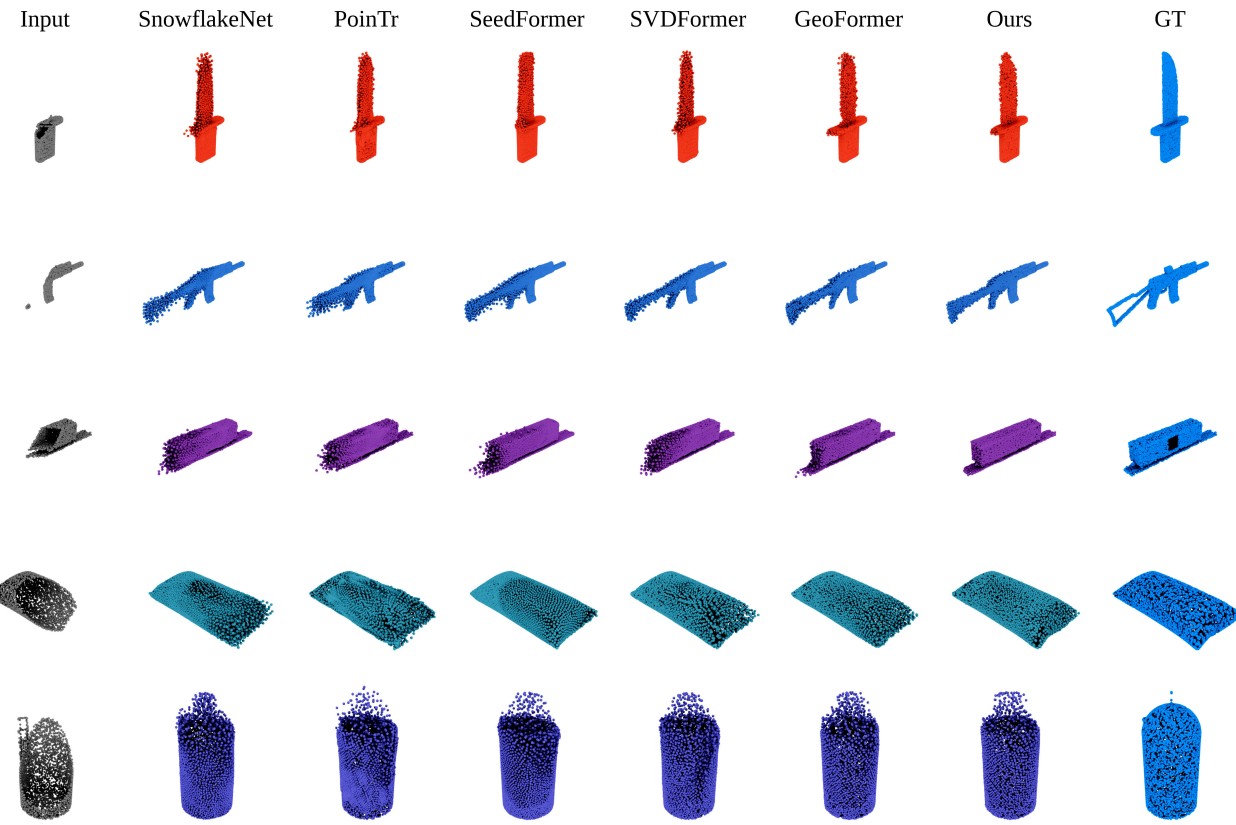

*Figure 10.* **Qualitative Results on ShapeNet-55 (Median Difficulty).** Comparisons of reconstruction quality on representative samples. SplAttN faithfully recovers details that are blurred by baselines.

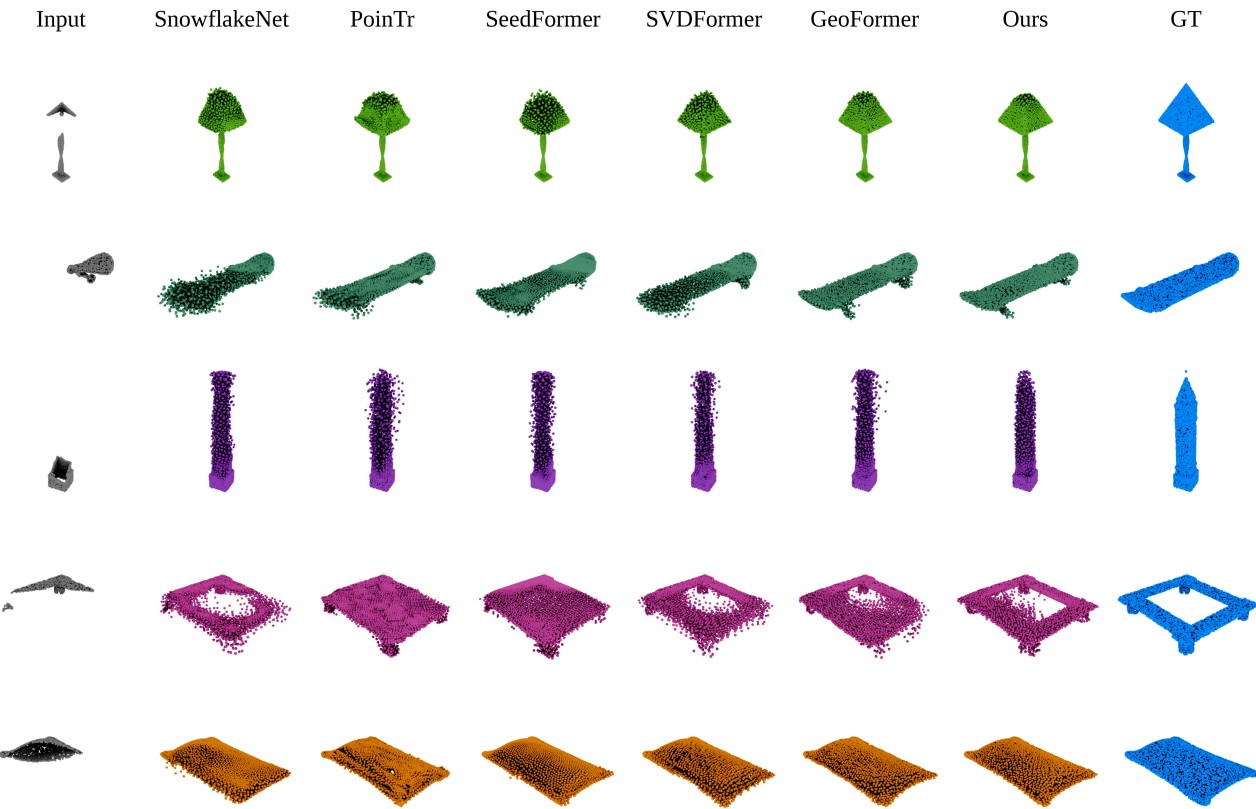

*Figure 11.* **Qualitative Results on ShapeNet-55 (Hard Difficulty).** Comparisons on challenging samples with significant missing geometry. Our method maintains structural integrity and input fidelity better than competitors.

## B. Detailed Performance on ShapeNet55/34

*Table 6.* **Detailed performance of SplAttN on ShapeNet55.** We report the L2 Chamfer Distance (CD, scaled by $10^3$) on Simple (S), Medium (M), and Hard (H) splits, along with the Average CD and F-Score@1% across all difficulties.

| Category | CD-S | CD-M | CD-H | **Avg-CD** | **Avg-F1** | Category | CD-S | CD-M | CD-H | **Avg-CD** | **Avg-F1** |
|---|---|---|---|---|---|---|---|---|---|---|---|
| Airplane | 0.19 | 0.30 | 0.52 | 0.33 | 0.708 | Knife | 0.15 | 0.28 | 0.44 | 0.29 | 0.822 |
| Bag | 0.34 | 0.51 | 0.94 | 0.60 | 0.517 | Lamp | 0.44 | 1.23 | 2.97 | 1.55 | 0.662 |
| Basket | 0.55 | 0.64 | 1.14 | 0.78 | 0.402 | Laptop | 0.26 | 0.28 | 0.43 | 0.32 | 0.536 |
| Bathtub | 0.43 | 0.64 | 1.16 | 0.74 | 0.462 | Mailbox | 0.23 | 0.58 | 1.76 | 0.86 | 0.654 |
| Bed | 0.50 | 0.75 | 1.52 | 0.92 | 0.446 | Microphone | 0.57 | 1.60 | 3.80 | 1.99 | 0.658 |
| Bench | 0.26 | 0.32 | 0.60 | 0.39 | 0.602 | Microwave | 0.53 | 0.64 | 1.29 | 0.82 | 0.407 |
| Birdhouse | 0.65 | 1.07 | 2.13 | 1.29 | 0.407 | Motorbike | 0.49 | 0.76 | 1.21 | 0.82 | 0.502 |
| Bookshelf | 0.50 | 0.70 | 1.31 | 0.84 | 0.418 | Mug | 0.64 | 0.88 | 1.75 | 1.09 | 0.371 |
| Bottle | 0.26 | 0.53 | 1.07 | 0.62 | 0.569 | Piano | 0.50 | 0.66 | 1.13 | 0.76 | 0.457 |
| Bowl | 0.46 | 0.50 | 0.90 | 0.62 | 0.409 | Pillow | 0.38 | 0.50 | 0.94 | 0.61 | 0.478 |
| Bus | 0.34 | 0.46 | 0.62 | 0.47 | 0.513 | Pistol | 0.33 | 0.54 | 0.86 | 0.58 | 0.613 |
| Cabinet | 0.46 | 0.53 | 0.84 | 0.61 | 0.402 | Pot | 0.71 | 1.08 | 1.96 | 1.25 | 0.413 |
| Camera | 0.70 | 1.41 | 2.81 | 1.64 | 0.417 | Printer | 0.49 | 0.88 | 1.91 | 1.09 | 0.439 |
| Can | 0.51 | 0.87 | 1.60 | 0.99 | 0.420 | Remote | 0.24 | 0.40 | 0.49 | 0.38 | 0.606 |
| Cap | 0.28 | 0.35 | 0.66 | 0.43 | 0.494 | Rifle | 0.23 | 0.37 | 0.64 | 0.41 | 0.767 |
| Car | 0.48 | 0.66 | 0.93 | 0.69 | 0.393 | Rocket | 0.15 | 0.37 | 0.80 | 0.44 | 0.779 |
| Cellphone | 0.26 | 0.30 | 0.38 | 0.31 | 0.561 | Skateboard | 0.17 | 0.26 | 0.37 | 0.27 | 0.687 |
| Chair | 0.32 | 0.51 | 1.11 | 0.65 | 0.528 | Sofa | 0.41 | 0.49 | 0.77 | 0.55 | 0.462 |
| Clock | 0.43 | 0.61 | 1.13 | 0.72 | 0.472 | Speaker | 0.53 | 0.80 | 1.54 | 0.96 | 0.422 |
| Dishwasher | 0.47 | 0.56 | 1.11 | 0.71 | 0.408 | Stove | 0.50 | 0.72 | 1.33 | 0.85 | 0.451 |
| Display | 0.33 | 0.48 | 0.90 | 0.57 | 0.512 | Table | 0.34 | 0.46 | 0.90 | 0.57 | 0.536 |
| Earphone | 0.51 | 0.81 | 3.04 | 1.45 | 0.499 | Telephone | 0.26 | 0.31 | 0.42 | 0.33 | 0.562 |
| Faucet | 0.52 | 1.00 | 2.30 | 1.27 | 0.617 | Tower | 0.38 | 0.74 | 1.53 | 0.88 | 0.559 |
| File Cabinet | 0.52 | 0.64 | 1.29 | 0.82 | 0.403 | Train | 0.35 | 0.52 | 0.82 | 0.56 | 0.545 |
| Guitar | 0.11 | 0.17 | 0.28 | 0.18 | 0.864 | Trash Bin | 0.59 | 0.83 | 1.46 | 0.96 | 0.364 |
| Helmet | 0.63 | 1.25 | 3.04 | 1.64 | 0.398 | Washer | 0.55 | 0.74 | 1.67 | 0.99 | 0.394 |
| Jar | 0.52 | 0.91 | 2.10 | 1.18 | 0.434 | Watercraft | 0.28 | 0.48 | 0.80 | 0.52 | 0.638 |
| Keyboard | 0.23 | 0.28 | 0.36 | 0.29 | 0.609 | | | | | | |

*Table 7.* **Detailed performance of SplAttN on ShapeNet-34/21 splits.** The left columns report results on 34 seen categories, while the right columns demonstrate zero-shot generalization on 21 unseen categories. We report L2 Chamfer Distance (CD, scaled by $10^3$) and F-Score@1%.

| | 34 Seen Categories | | | | | | 21 Unseen Categories | | | | |
|---|---|---|---|---|---|---|---|---|---|---|---|
| Category | CD-E | CD-M | CD-H | **Avg-CD** | **Avg-F1** | Category | CD-E | CD-M | CD-H | **Avg-CD** | **Avg-F1** |
| Airplane | 0.32 | 0.52 | 1.00 | 0.62 | 0.569 | Bag | 0.50 | 0.92 | 2.03 | 1.15 | 0.483 |
| Bathtub | 0.32 | 0.59 | 0.93 | 0.61 | 0.549 | Basket | 0.56 | 1.09 | 2.27 | 1.31 | 0.467 |
| Bed | 0.38 | 0.57 | 1.00 | 0.65 | 0.526 | Birdhouse | 0.49 | 0.93 | 2.03 | 1.15 | 0.482 |
| Bench | 0.35 | 0.52 | 0.89 | 0.59 | 0.531 | Bowl | 0.49 | 0.84 | 1.80 | 1.04 | 0.497 |
| Bookshelf | 0.38 | 0.57 | 1.24 | 0.73 | 0.520 | Camera | 0.47 | 0.81 | 1.77 | 1.02 | 0.490 |
| Bottle | 0.37 | 0.59 | 1.18 | 0.71 | 0.520 | Can | 0.46 | 0.91 | 2.19 | 1.18 | 0.478 |
| Bus | 0.39 | 0.62 | 1.07 | 0.69 | 0.527 | Cap | 0.54 | 1.02 | 2.45 | 1.34 | 0.487 |
| Cabinet | 0.46 | 0.65 | 1.19 | 0.77 | 0.498 | Dishwasher | 0.55 | 0.97 | 2.22 | 1.24 | 0.459 |
| Chair | 0.32 | 0.47 | 0.89 | 0.56 | 0.559 | Earphone | 0.53 | 1.02 | 2.37 | 1.31 | 0.489 |
| Clock | 0.37 | 0.61 | 1.07 | 0.68 | 0.538 | File Cabinet | 0.46 | 0.65 | 1.19 | 0.77 | 0.483 |
| Display | 0.37 | 0.57 | 1.11 | 0.68 | 0.528 | Keyboard | 0.48 | 0.90 | 2.15 | 1.18 | 0.476 |
| Faucet | 0.36 | 0.52 | 0.89 | 0.59 | 0.537 | Mailbox | 0.49 | 0.83 | 2.11 | 1.14 | 0.463 |
| Guitar | 0.37 | 0.53 | 1.03 | 0.64 | 0.541 | Microwave | 0.59 | 1.04 | 2.57 | 1.40 | 0.483 |
| Jar | 0.37 | 0.59 | 1.09 | 0.68 | 0.536 | Motorbike | 0.49 | 0.58 | 1.19 | 0.75 | 0.473 |
| Knife | 0.40 | 0.56 | 1.07 | 0.68 | 0.533 | Pillow | 0.53 | 1.08 | 2.47 | 1.36 | 0.467 |
| Lamp | 0.38 | 0.54 | 0.99 | 0.64 | 0.546 | Printer | 0.56 | 1.06 | 2.43 | 1.35 | 0.494 |
| Laptop | 0.45 | 0.64 | 1.30 | 0.79 | 0.503 | Remote | 0.24 | 0.40 | 0.50 | 0.38 | 0.572 |
| Microphone | 0.54 | 0.99 | 2.45 | 1.33 | 0.475 | Rocket | 0.52 | 1.01 | 2.06 | 1.20 | 0.485 |
| Mug | 0.45 | 0.58 | 1.19 | 0.74 | 0.490 | Skateboard | 0.56 | 0.98 | 2.31 | 1.29 | 0.469 |
| Piano | 0.41 | 0.59 | 1.15 | 0.72 | 0.521 | | | | | | |
| Pistol | 0.49 | 0.67 | 1.31 | 0.82 | 0.482 | | | | | | |
| Pot | 0.41 | 0.56 | 1.03 | 0.66 | 0.521 | | | | | | |
| Rifle | 0.37 | 0.54 | 0.89 | 0.60 | 0.571 | | | | | | |
| Sofa | 0.32 | 0.46 | 0.80 | 0.53 | 0.567 | | | | | | |
| Speaker | 0.35 | 0.51 | 0.98 | 0.61 | 0.552 | | | | | | |
| Stove | 0.46 | 0.65 | 1.25 | 0.79 | 0.476 | | | | | | |
| Table | 0.33 | 0.48 | 0.89 | 0.57 | 0.552 | | | | | | |
| Telephone | 0.37 | 0.56 | 1.08 | 0.67 | 0.521 | | | | | | |
| Tower | 0.57 | 0.94 | 1.95 | 1.15 | 0.496 | | | | | | |
| Train | 0.39 | 0.59 | 1.01 | 0.66 | 0.511 | | | | | | |
| Trash Bin | 0.38 | 0.56 | 1.01 | 0.65 | 0.521 | | | | | | |

## C. Additional Theoretical Analysis

In this appendix, we provide a theoretical analysis showing that our density estimation framework, motivated by the practical need to avoid extremely sparse image-plane support, implicitly and approximately maximizes the mutual information between geometric and visual modalities under an idealized model. We also formally define the Cross-Modal Information Throughput metric.

### C.1. PMI-Style Interpretation of Density-Based Splatting

We provide a concise information-theoretic interpretation of density-based splatting. This section is intended as an explanatory perspective rather than an additional explicit optimization objective.

Let $q \in \Omega$ denote a continuous image-plane query location, corresponding to the spatial query used in Eq. (6). The mutual information between the sparse geometric observation $\mathcal{P}_{in}$ and $q$ can be written as

$$I(\mathcal{P}_{in}; q) = H(q) - H(q \mid \mathcal{P}_{in}). \tag{12}$$

When the marginal query distribution $P(q)$ is treated as fixed, reducing the conditional uncertainty of $q$ given $\mathcal{P}_{in}$ corresponds to increasing their mutual information.

Hard projection induces a degenerate geometry-conditioned distribution:

$$P_{\text{hard}}(q \mid \mathcal{P}_{in}) = \frac{1}{N} \sum_{p \in \mathcal{P}_{in}} \delta(q - \pi(p)). \tag{13}$$

Its support consists only of isolated projected pixels and has zero measure in the continuous image plane. Therefore, off-support visual queries receive no smooth local response, which limits gradient-based cross-modal alignment.

By contrast, Gaussian splatting replaces the Dirac measure with a smooth kernel:

$$P_{\text{soft}}(q \mid \mathcal{P}_{in}) = \frac{1}{Z} \sum_{p \in \mathcal{P}_{in}} \alpha_p G(q; \pi(p), \sigma), \tag{14}$$

where $G$ is the Gaussian kernel, $\alpha_p \geq 0$ denotes the positive contribution weight of point $p$, and $Z$ normalizes the weighted mixture over $\Omega$. In our implementation, $\alpha_p$ corresponds to the depth-aware contribution term used in Eq. (7), and this density induces the normalized aggregation weights for the feature expectation in Eq. (6).

Under this density view, the point-wise mutual information can be expressed as

$$\text{PMI}(\mathcal{P}_{in}, q) = \log \frac{P_{\text{soft}}(q \mid \mathcal{P}_{in})}{P(q)}. \tag{15}$$

Since $P(q)$ is fixed with respect to the model parameters, assigning higher density to compatible query regions can be interpreted as encouraging high-PMI point-wise correspondences. Thus, our splatting module does not explicitly maximize mutual information through a separate NLL or contrastive loss; instead, it provides a differentiable density support on which cross-attention can more effectively establish visual-geometric dependency.

### C.2. Multi-Channel Information Capacity Measurement

While the derivation above proves the optimization objective, quantifying the resulting information density requires careful handling of multi-dimensional features. Standard analysis often treats the projected feature map as a scalar field. However, geometric projections encode spatial information into distinct orthogonal channels $c \in \{1, \ldots, C\}$.

We argue that averaging these channels underestimates the true information capacity. Formally, the total entropy of a multi-channel feature map $\mathbf{V}$ over a local region $\Omega$ is approximated by the sum of marginal entropies, assuming channel orthogonality in the geometric basis:

$$H(\mathbf{V}_\Omega) \approx \sum_{c=1}^{C} H(\mathbf{V}_{\Omega,c}) \tag{16}$$

where $H(\mathbf{V}_{\Omega,c})$ denotes the Shannon entropy of the $c$-th channel. Standard hard projection yields a support set $\mathcal{S}_{hard}$ with measure zero across all channels simultaneously. In contrast, SplAttN expands the support $\mathcal{S}_{soft}$ within each channel independently via the Gaussian kernel $\mathcal{G}(\cdot;\sigma)$, effectively maximizing the joint information density passed to the visual backbone.

### C.3. Cross-Modal Information Throughput

To provide a holistic measure of the information flow passed from the geometric encoder to the visual backbone, we introduce the Cross-Modal Information Throughput (CMIT). While Entropy ($H$) measures the information density per active region and Coverage ($C$) measures the spatial extent of valid signals, neither metric alone captures the total effective signal yield. For instance, a projection could have high entropy but exist at only a single pixel, or have 100% coverage but contain uniform noise.

We define CMIT as the product of the channel-aware entropy and the spatial coverage ratio:

$$\text{CMIT}(\mathbf{V}) = H(\mathbf{V}) \times C(\mathbf{V}) \tag{17}$$

This metric serves as a proxy for the Total Information Yield. A high CMIT indicates that the connection function successfully preserves the complexity of the input geometry while distributing it effectively across the latent visual manifold. As shown in our KITTI experiments, SplAttN achieves a CMIT orders of magnitude higher than baseline methods, validating its ability to prevent entropy collapse and maximize the learnable connection.

## D. Metric Definitions and Implementation Details

### D.1. Fidelity Distance and Minimum Matching Distance

To evaluate the reconstruction quality on the real-world KITTI benchmark, we follow standard protocols (Yu et al., 2021) and report these two metrics.

**Fidelity Distance.** FD quantifies how faithfully the completed point cloud $\mathcal{P}_{out}$ preserves the observed geometry from the partial input $\mathcal{P}_{in}$. It is computed as the one-sided Chamfer Distance:

$$\text{FD}(\mathcal{P}_{in}, \mathcal{P}_{out}) = \frac{1}{|\mathcal{P}_{in}|} \sum_{p \in \mathcal{P}_{in}} \min_{q \in \mathcal{P}_{out}} \|p - q\|_2 \tag{18}$$

A lower value implies better adherence to the raw sensor data, ensuring the model does not hallucinate structures that contradict the input.

**Minimum Matching Distance.** MMD measures the output plausibility by finding the closest shape in a reference set $\mathcal{S}_{ref}$, which is typically the ShapeNet-Cars training set.

$$\text{MMD}(\mathcal{P}_{out}, \mathcal{S}_{ref}) = \min_{G \in \mathcal{S}_{ref}} \text{CD}(\mathcal{P}_{out}, G) \tag{19}$$

However, we argue that the MMD calculation is both *cumbersome* and *meaningless* in this Sim-to-Real setting. First, finding the nearest neighbor requires traversing the entire ShapeNet-Cars dataset for every single test sample, which incurs prohibitive computational costs. Second, forcing a match between real-world LiDAR scans and holistic synthetic templates introduces a fundamental domain bias. This discrepancy arises because LiDAR data exhibits ray-like sparsity and sensor noise, rendering the metric unreliable for assessing true reconstruction fidelity.

### D.2. Semantic Consistency Score

To assess whether the reconstructed point clouds preserve semantically recognizable structures, we define the Semantic Consistency Score (SCS). We utilize a pre-trained classification network, DGCNN (Wang et al., 2019), as an oracle evaluator.

Let $\mathcal{F}_{\text{DGCNN}} : \mathbb{R}^{N \times 3} \to [0, 1]$ denote the classifier mapping a point cloud to the confidence score of its ground-truth category. The SCS is defined as:

$$\text{SCS} = \mathcal{F}_{\text{DGCNN}}(\mathcal{P}_{\text{completed}}) \tag{20}$$

A higher SCS indicates that the completed shape possesses high-fidelity semantic features recognizable by a standard 3D classifier.

### D.3. KITTI Normalization Protocol

To mitigate the domain gap between the real-world KITTI scans and the synthetic ShapeNet training data, we apply a standardized pose normalization to the KITTI car instances before evaluation. Given a raw point cloud $\mathcal{P}_{\text{raw}}$ and its annotated 3D bounding box $\mathcal{B}$, the normalization $\mathcal{T}(\cdot)$ consists of four steps:

1. **Centering:** We compute the geometric center of the bounding box $c_{\text{bbox}} = (\mathcal{B}_{\text{min}} + \mathcal{B}_{\text{max}})/2$ and translate the point cloud to the origin:

$$p' = p - c_{\text{bbox}}, \quad \forall p \in \mathcal{P}_{\text{raw}} \tag{21}$$

2. **Rotation Alignment:** We calculate the yaw angle $\theta$ of the bounding box to determine the object's orientation. We then apply a rotation matrix $\mathbf{R}_z(-\theta)$ to align the car's heading with the canonical axis:

$$p'' = \mathbf{R}_z(-\theta) \cdot p' \tag{22}$$

3. **Canonical Scaling:** We uniformly scale the point cloud using the length of the bounding box (main axis) as the normalization factor $s$:

$$p''' = \frac{p''}{s} \tag{23}$$

4. **Coordinate Transformation:** Finally, we permute the axes to match the ShapeNet coordinate system convention (up-axis alignment), transforming $(x, y, z)$ to $(x, z, y)$.

This protocol ensures that the zero-shot evaluation on KITTI strictly measures the reconstruction capability rather than robustness to arbitrary pose variations.

## E. Comparison on Computational Cost

Table 8 reports the computational cost of each method measured on a single NVIDIA RTX 3090 over the PCN test set. Despite having a larger parameter count, SplAttN achieves competitive inference latency and GPU memory usage relative to methods of similar complexity, while delivering superior reconstruction quality.

*Table 8.* Computational cost comparison. CD-Avg is reported from Table 1. Params, MACs, latency, and GPU memory are measured on a single NVIDIA RTX 3090 over the PCN test set with batch size 1.

| Method | CD-Avg ↓ | Params ↓ | MACs ↓ | Latency ↓ | GPU Mem ↓ |
|---|---|---|---|---|---|
| PCN (Yuan et al., 2018) | 9.64 | 6.86M | 14.71G | 1.90ms | 0.17GB |
| SnowflakeNet (Xiang et al., 2021) | 7.21 | 19.32M | 10.32G | 10.97ms | 0.18GB |
| PoinTr (Yu et al., 2021) | 8.38 | 42.50M | 11.61G | 11.05ms | 0.23GB |
| SeedFormer (Zhou et al., 2022) | 6.74 | 3.31M | 53.76G | 40.39ms | 0.48GB |
| AdaPoinTr (Yu et al., 2023) | 6.53 | 32.49M | 15.09G | 11.62ms | 0.17GB |
| SVDFormer (Zhu et al., 2023b) | 6.54 | 58.09M | 39.26G | 31.72ms | 0.55GB |
| GeoFormer (Yu et al., 2024) | 6.42 | 58.23M | 39.38G | 31.06ms | 0.68GB |
| Ours | 6.36 | 65.89M | 38.26G | 40.75ms | 0.58GB |

## F. Additional Entropy Collapse Analysis

In this section, we present additional visualization results for the Entropy Collapse analysis on the PCN dataset. Figures 12 and 13 compare the feature representations of Hard Depth, Hard CCM, and our proposed Soft Splatted CCM. As shown, our method effectively mitigates entropy collapse, maintaining high feature coverage across different samples.

**Entropy Collapse Analysis: Hard Projection vs Gaussian Splatting**

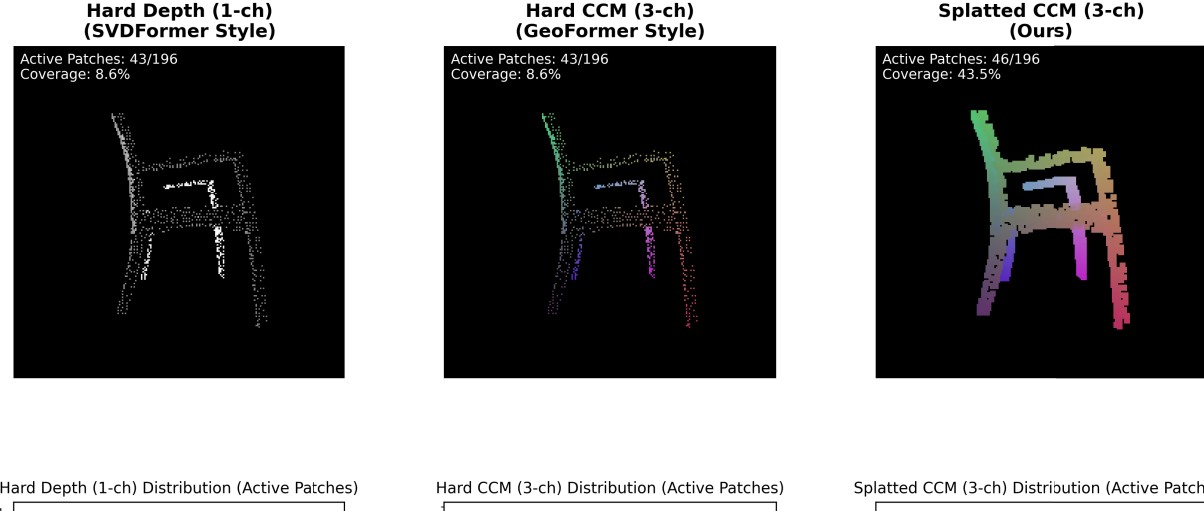

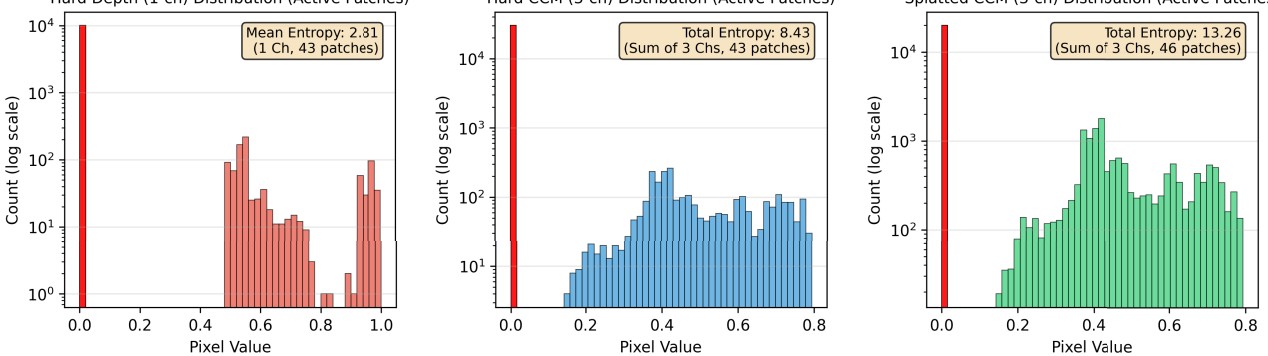

*Hard projections produce sparse feature maps within active regions, causing information loss.*
*Gaussian splatting (Ours) distributes information smoothly within each patch, preserving spatial continuity.*

*Figure 12.* **Entropy Analysis - Sample 1.** Our method produces dense feature maps compared to sparse baselines.

**Entropy Collapse Analysis: Hard Projection vs Gaussian Splatting**

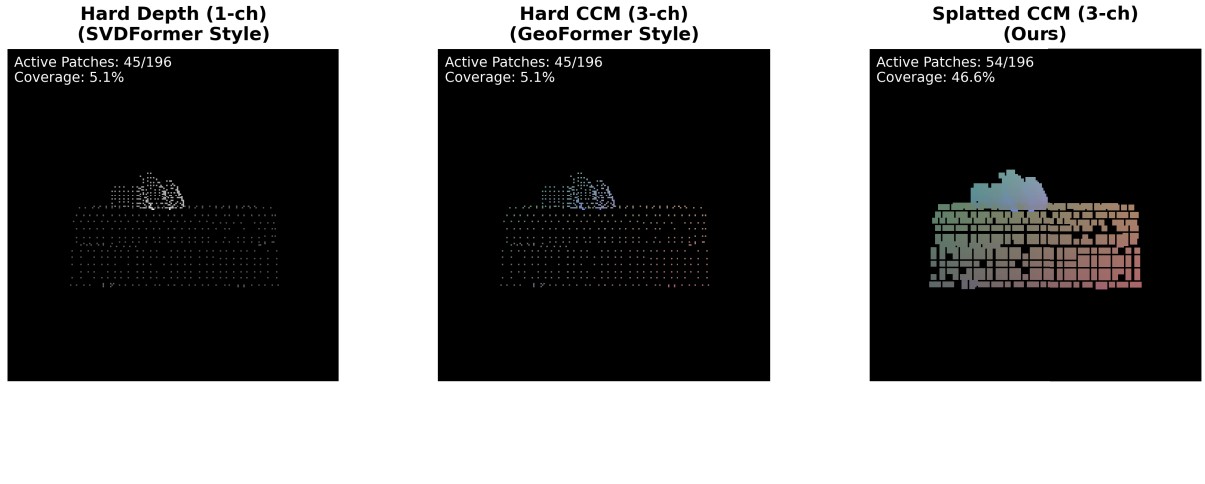

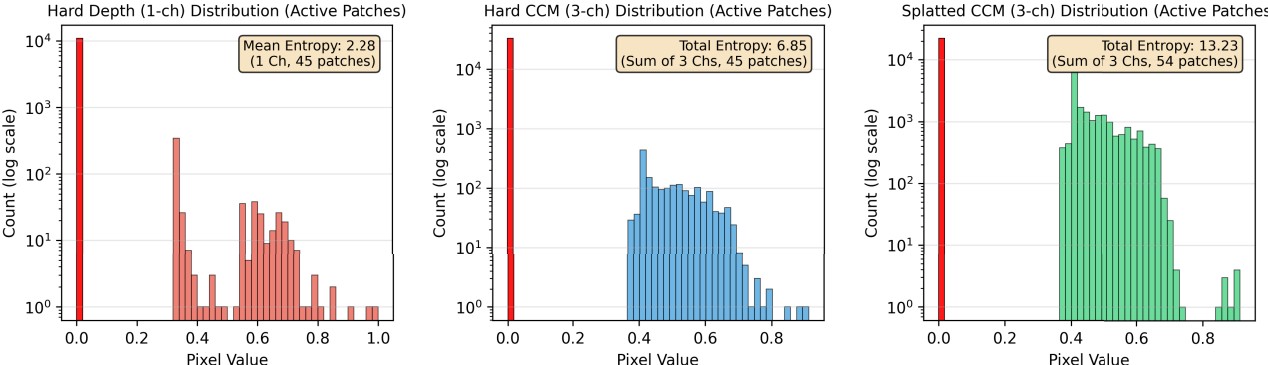

*Hard projections produce sparse feature maps within active regions, causing information loss.*
*Gaussian splatting (Ours) distributes information smoothly within each patch, preserving spatial continuity.*

*Figure 13.* **Entropy Analysis - Sample 2.** Histogram analysis demonstrates the broader value distribution of our method.

## G. Additional KITTI Qualitative Results

We present additional qualitative comparisons on the real-world KITTI dataset (Geiger et al., 2013) in Figure 14. KITTI serves as a challenging stress test for multi-modal point cloud completion methods, as its scans exhibit significantly sparser and noisier distributions compared to synthetic benchmarks. The visualized reconstructions are consistent with the trends reflected by our Semantic Consistency Score (SCS) metric, suggesting that SCS captures meaningful differences in cross-modal dependency that align with perceptual quality observed in the real-world setting.

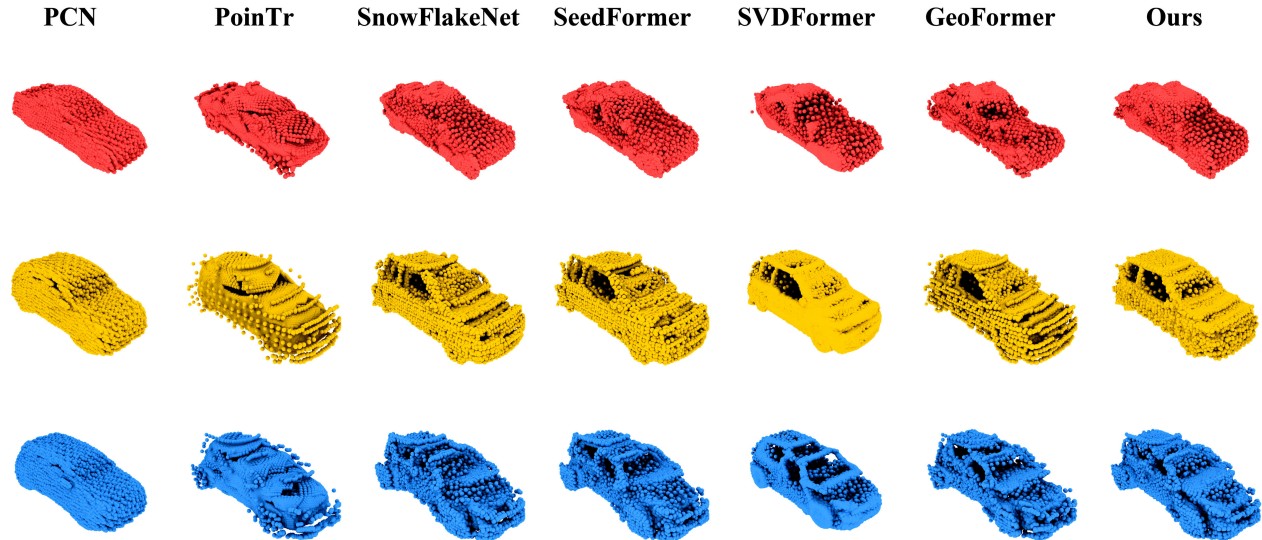

*Figure 14.* **Qualitative Results on KITTI.** Comparisons of point cloud completion on real-world scans. The visual differences across methods are consistent with the rankings produced by our Semantic Consistency Score (SCS) metric.

## H. Additional KITTI Robustness Analysis

To further investigate the performance trade-off discussed in the main text, we provide a detailed visualization of the intermediate feature representations in Figure 15. This comparison highlights the impact of projection strategies on information retention under domain shift.

As shown in the top row of Figure 15, the raw LiDAR scans from KITTI are characterized by extreme sparsity. When utilizing deterministic Hard Projection (used in SVDFormer and GeoFormer), the resulting feature maps suffer from severe information loss, with active pixel coverage dropping below 10% (e.g., 5.3% in the Front View). This sparsity implies that the visual backbone receives limited gradient support from the geometric input, potentially forcing the model to rely heavily on learned priors.

In contrast, our SplAttN utilizes Differentiable Gaussian Splatting to estimate a continuous density field. As quantified in Figure 15, this mechanism significantly expands the valid information support, increasing the feature coverage by approximately $4.3\times$ (e.g., reaching 25.7% in the Top View). The visualization confirms that our method effectively preserves the spatial structure of the sparse input, bridging the modality gap without discarding the unique geometric signatures of the real-world sensor data.

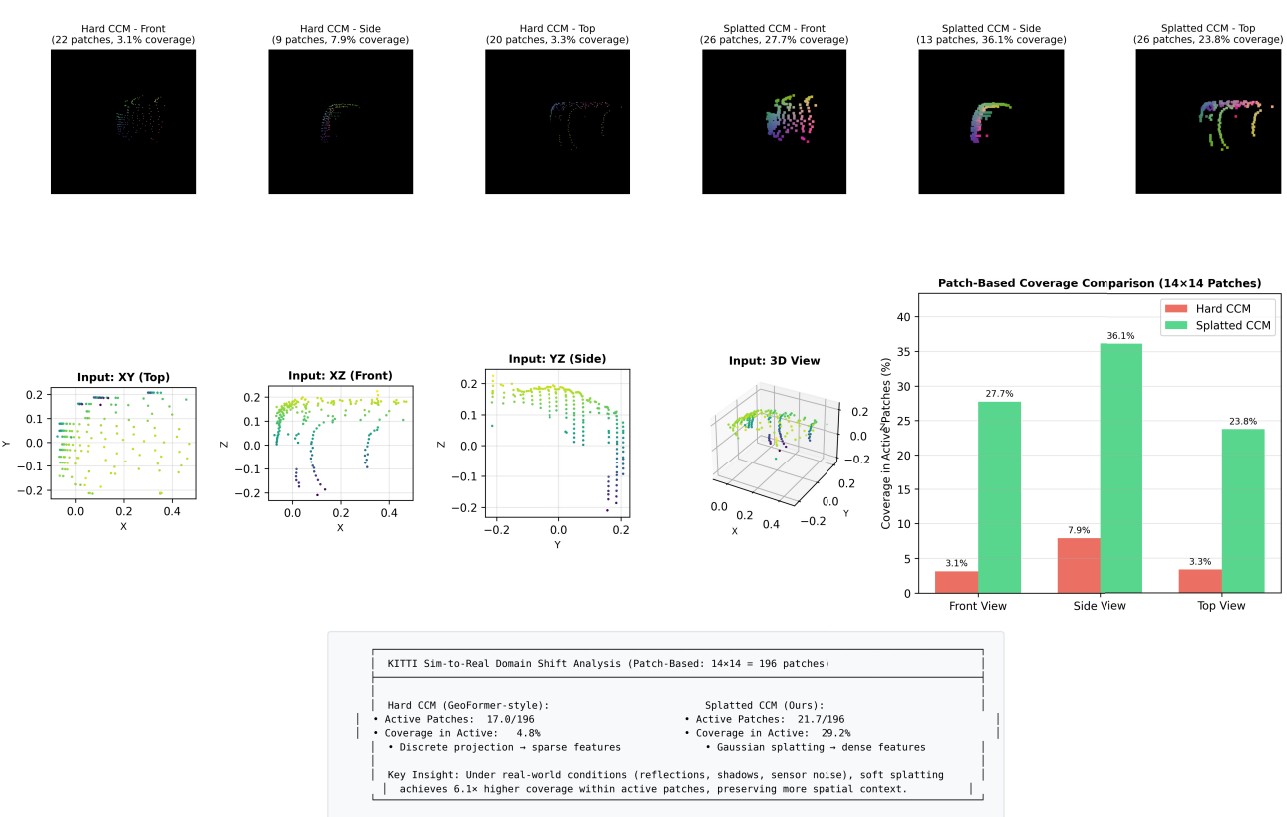

*Figure 15.* **KITTI Robustness - Sample 1.** Three-view feature comparison under sim-to-real domain shift.

*Figure 16.* **KITTI Robustness - Sample 2.** Visualization of point cloud projection and feature map coverage.

