# OpenReview forum: "SplAttN: Bridging 2D and 3D with Gaussian Soft Splatting and Attention for Point Cloud Completion"
_ICML.cc/2026/Conference — ICML 2026 spotlight_

### Official Review · Reviewer_LT9w · 2026-03-06

**Soundness:** 3
**Presentation:** 2
**Significance:** 3
**Originality:** 3
**Overall Recommendation:** 4
**Confidence:** 3

**Summary:**

This paper proposes a point cloud completion method that leverages projected 2D images to assist the completion of 3D point clouds. To bridge the gap between the 3D geometry and 2D visual information, the paper introduces a Gaussian splatting-based mechanism for cross-modal interaction. Based on this design, the authors present a supervised framework that achieves strong empirical performance.

Furthermore, to analyze the model’s dependency on visual information, the authors conduct additional experiments on the KITTI dataset. The results suggest that the proposed approach exhibits a stronger reliance on 2D image cues compared to simple projection-based baselines, indicating that the method effectively utilizes multimodal information.

**Compliance With Llm Reviewing Policy:**

Affirmed.

**Final Justification:**

The rebuttal addressed my concerns, and I am now confident in my evaluation.

**Key Questions For Authors:**

- How many projection views are used in practice? In Figure 3, multiple splatted projections appear as inputs. Are all of these views used simultaneously during training and inference, or are they sampled in some way?

- In Equation (6), the term $f_k$ appears without a clear definition. Could the authors clarify what feature representation $f_k$ corresponds to and how it is obtained?

- The proposed framework appears to assume a known projection between the 3D point cloud and the 2D image. How robust is the method when the projection is noisy or slightly inaccurate?

**Limitations:**

- While the proposed Gaussian splatting-based bridge is effective, the overall framework remains supervised and relies on paired 3D point clouds and 2D image information during training. This may limit its applicability in settings where such aligned annotations are difficult to obtain.

- Although the paper demonstrates that the method makes better use of visual information than simple projection-based baselines, the extent to which the approach generalizes under severe image degradation, strong viewpoint mismatch, or inaccurate projection remains unclear.

- The analysis of the proposed multimodal interaction could be further strengthened. In particular, additional studies on the number of projections and the effect of the Gaussian kernel parameter would help clarify which components are most responsible for the observed gains.

**Strengths And Weaknesses:**

## Strengths

- The method effectively captures both global and local characteristics of point clouds by leveraging multi-modal information.
- The use of Gaussian splatting appears to be a well-motivated design choice, and the ablation study demonstrates its effectiveness.
- The method achieves strong empirical performance across benchmarks.
- The paper conducts dependency tests on the multimodal inputs, providing evidence that the model indeed utilizes visual information as intended.

## Weaknesses

- It would be helpful to provide qualitative reconstruction results for the KITTI dataset. Since the quality and alignment of the 2D images in KITTI may be limited, visual examples could clarify how the proposed method behaves in such scenarios.
- The paper would benefit from a clearer explanation of how many projections are actually used. In Figure 3, multiple splatted views appear as inputs; it would be useful to clarify whether all of them are used simultaneously. Additionally, an analysis showing how performance varies with the number of projections would strengthen the paper.
- The notation could be clarified and made more consistent throughout the paper.
  - The terms $F_{\text{vis}}$ and $F_{img}$ appear in Figure 1 and Figure 3 respectively; unifying the notation would improve readability.
  - In Equation (6), the term $f_k$ appears without a clear definition and should be explicitly explained. Similarly, the term $z_k$ in Equation (7) requires clarification.
  - The value used for the Gaussian kernel parameter $\sigma$ is not specified and should be reported.

---

> ### Author Rebuttal · Authors · 2026-03-28
>
> We thank the reviewer for the positive assessment and constructive suggestions. We are encouraged that the reviewer recognizes the value of the Gaussian splatting bridge, the multimodal dependency analysis, and the strong empirical performance.
>
> 1. On qualitative KITTI results
> We thank the reviewer for this helpful suggestion. We agree that qualitative KITTI reconstructions are useful for understanding model behavior under imperfect image quality and alignment. To make this more transparent, we have provided additional qualitative results in our anonymous repository: https://anonymous.4open.science/r/Additional_results-4305
> These examples complement the quantitative analysis and give a clearer picture of how the model behaves in realistic KITTI scenarios.
>
> 2. On the number of projection views
> We use 3 fixed projection views simultaneously during both training and inference; they are not sampled. We chose this setting partly following the analysis in SVDFormer, where among 1-view, 3-view, and 6-view settings, 3 views gave the best balance between performance and computation. We therefore adopted the same setting as a practical default. At the same time, we agree that, after changing the projection mechanism to Gaussian splatting, it is worthwhile to revisit this analysis under our framework. We thank the reviewer for this suggestion, and a systematic study of the number of projections will be included in future work.
>
> 3. On Eq. (6), Eq. (7), notation and Gaussian kernel size
> We thank the reviewer for these careful comments on notation and presentation. We agree that the notation can be made more consistent throughout the paper.
> For Figures 1 and 3, we will further unify the notation to improve readability. We suspect that the reviewer may have meant $F_{\mathrm{geo}}$ rather than $F_{\mathrm{img}}$, but in any case we appreciate the observation and will make the figure labeling clearer in the revision.
> For Eq. (6), $f_k$ denotes the feature attached to the $k$-th projected primitive; in our current CCM implementation, this is concretely instantiated as a 3-channel pseudo-color derived from normalized 3D coordinates. In Eq. (7), $z_k$ denotes the depth of the $k$-th projected primitive used in the depth-aware weighting term. We will define these symbols more explicitly in the main text.
> For the Gaussian kernel parameter, the kernel size is set to 4, which is already stated explicitly in the *Implementation and Metrics* subsection. We thank the reviewer for noticing this and we will ensure the notation is thoroughly consistent throughout the final version.
>
> 4. On robustness to inaccurate projection and calibration noise
> The reviewer rightly points out that robustness to noisy or slightly inaccurate projection is an important question. In the current paper, our main goal is to validate the proposed cross-modal bridge under the standard object-level completion setting, where the projection is assumed known. At the same time, the additional qualitative KITTI results in our anonymous repository suggest that the method remains reasonably stable under realistic image quality and alignment imperfections:
> https://anonymous.4open.science/r/Additional_results-4305
> That said, we agree that this does not replace a controlled perturbation study. A systematic analysis under calibration noise or projection perturbation would further strengthen the paper, and we will include this as an important direction for future work.
>
> 5. On supervised learning and paired annotations
> We appreciate the reviewer raising this point. While our framework is supervised, fully supervised learning remains the standard paradigm in most point cloud completion research. Our method only requires partial point clouds during inference. Compared to multi source models that demand paired 2D images at test time, our approach does not impose any additional annotation or data gathering burden during real world deployment.
>
> 6. On generalization under severe degradation and mismatch
> We agree this is a valuable research direction. Our primary objective is to establish the cross modal bridge under standard settings. We therefore do not claim strong generalization under extreme out of distribution scenarios and view this as an important next step.
>
> 7. On further analysis of multimodal interaction
> We agree that exploring these interaction components is very valuable. But as mentioned in the second point, studying factors like projection counts and kernel parameters requires a dedicated evaluation. We plan to explore this thoroughly in future work as it naturally extends the scope of the current paper.
>
> We thank the reviewer again for the constructive suggestions.

---

> > ### Author Rebuttal · Reviewer_LT9w · 2026-04-03
> >
> > My concerns have been adequately addressed. I believe the paper would benefit from incorporating the results obtained during the rebuttal process into the revised manuscript. I will maintain my positive score.

---

> > > ### Author Response · Authors · 2026-04-03
> > >
> > > We appreciate the reviewer's recognition of our rebuttal and are pleased to have addressed your concerns. We have fully adopted your suggestions and will carefully revise the manuscript. We hope our final work will be a valuable contribution to the community.

---

### Official Review · Reviewer_J3Ch · 2026-03-12

**Soundness:** 3
**Presentation:** 3
**Significance:** 2
**Originality:** 3
**Overall Recommendation:** 4
**Confidence:** 2

**Summary:**

This paper proposes SplAttN, a multimodal point cloud completion framework that replaces hard, discrete projection from 3D to 2D with a differentiable Gaussian soft splatting mechanism, aiming to maintain gradient flow and strengthen the learnable connection between geometric and visual modalities. The authors motivate this with a theoretical argument that hard projection induces cross-modal “entropy collapse” and show that soft density estimation expands support and yields non-vanishing gradients; they further couple this with an active attention scheme (geometric queries over densified visual keys/values) and a hybrid global-local encoder/decoder. Experiments on PCN and ShapeNet-55/34 show state-of-the-art or near state-of-the-art results, and a counterfactual study on KITTI using a Semantic Consistency Score suggests SplAttN relies more on visual cues than competing multimodal baselines.

**Compliance With Llm Reviewing Policy:**

Affirmed.

**Final Justification:**

Most of my concerns have been resolved.

**Key Questions For Authors:**

1. Are the image inputs on PCN/ShapeNet external RGB images or self-rendered views from the partial point cloud? How many views are used, how are camera poses chosen, and are textures/colors available or synthesized?

2. How does the model perform when images are absent at inference (modality dropout)? Please report quantitative results and compare to unimodal backbones to substantiate the reliance vs robustness trade-off.

**Limitations:**

no, please refer to the weakness.

**Strengths And Weaknesses:**

**Strengths**

1.Recasts point-to-image projection as continuous density estimation via Gaussian soft splatting, making the cross-modal mapping differentiable and potentially more informative than sparse, hard projection.

2.The “active attention” design where geometric tokens query a densified visual field is a clean, principled fusion mechanism aligned with the motivation of maximizing point-wise mutual information.

3.Introduces a depth-aware weighting (soft Z-buffer) and a hybrid local-global geometric tokenizer, integrating curvature-aware EdgeConv and transformer-based global reasoning.

4.The paper is generally well-written with clear architectural figures and a coherent narrative from theory to implementation.

**Weakness**

1.The claim of “maximizing PMI” is not operationalized in the loss or training objective; Eq. (8) implements cross-attention, but no explicit MI estimator or surrogate objective is used. The theoretical link remains suggestive rather than concrete.

2.The argument that hard projection yields vanishing gradients is overstated: many pipelines use differentiable image sampling (e.g., bilinear grid sampling/RoIAlign) that provide gradients even with deterministic projection; the comparison baseline should include such differentiable samplers.

3.The depth-aware occlusion model (1/z) is a heuristic that can be unstable near the camera; no analysis of robustness to σ, ε, or camera calibration errors is provided.

4.KITTI evaluation relies on a Semantic Consistency Score from a pre-trained classifier; details about the classifier, domain alignment, and its correlation with geometric quality are limited. No geometric metrics (e.g., partial-to-complete CD with available proxies) are reported for KITTI, leaving external validity uncertain.

5.Missing or limited comparisons to recent multimodal completion methods such as PointSea and MGPC that also propose cross-modal fusion strategies; this weakens the empirical claim of superiority in the multimodal regime.

6.Lack of runtime, memory, and computational overhead analysis for splatting vs hard projection or bilinear sampling—important for practical adoption.

---

> ### Author Rebuttal · Authors · 2026-03-28
>
> We thank the reviewer for the careful reading and thoughtful feedback. We are encouraged that the reviewer recognizes the novelty of continuous density estimation for projection, the active-attention design, and the hybrid geometric tokenizer.
>
> 1. On the “maximizing PMI” claim
> As the reviewer correctly observed Eq. (8) is not an explicit MI estimator, and the training loss does not directly optimize a standalone MI objective. Our intention is therefore not to claim exact MI optimization at the loss-function level. Rather, PMI is used as a theoretical lens to motivate the GS-Bridge design: soft density estimation expands valid support and enables geometric queries to retrieve from a denser visual field, thereby strengthening cross-modal dependence during feature interaction. We will revise the wording to make this distinction clearer.
>
> 2. On hard projection and differentiable samplers
> We acknowledge that our original wording can sound too broad. Our critique is specifically directed at the sparse hard projection used in prior multimodal point cloud completion pipelines. Bilinear sampling and RoIAlign can propagate gradients, but they operate in a different regime: they sample from an already available dense feature field, whereas our focus is on the severe support sparsity introduced at the 3D-to-2D projection stage itself. To the best of our knowledge, current multimodal point cloud completion baselines do not use these differentiable samplers as their 3D-to-2D projection mechanism, so a direct like-for-like comparison is not straightforward. We will revise the wording to make this distinction clearer.
>
> 3. On the depth-aware weighting
> We agree that the $(z+\epsilon)^{-1}$ term is a practical heuristic rather than a claim of optimal physical modeling. In our framework, it serves as a simple soft-occlusion prior to favor closer projected primitives, while $\epsilon$ is used for numerical stability. Our main contribution is the Gaussian splatting-based cross-modal bridge itself, rather than an exhaustive study of the best depth-weighting function. We also share the reviewer's view that robustness to $\sigma$, $\epsilon$, and calibration noise would further strengthen the paper, and we will clarify this limitation and include additional analysis where space permits.
>
> 4. On KITTI evaluation and metric
> It is indeed true that no single metric is fully satisfactory on KITTI. Standard paired CD is not directly applicable because KITTI does not provide complete ground-truth shapes. Prior work therefore often relied on proxy geometric statistics such as FD or MMD, which also have limitations under domain shift and do not directly capture multimodal reliance. Our use of SCS is thus not intended as a replacement for geometric evaluation, but as a counterfactual diagnostic tailored to the KITTI experiment: whether removing the image branch causes consistent semantic degradation. We agree that SCS is imperfect, and we will clarify this intended scope more explicitly.
>
> 5. On comparisons to  MGPC and PointSea
> For MGPC, it is a very recent concurrent work and was not a stable comparison target during our submission cycle: it was first posted on Jan. 7, 2026, and the current v2 was revised and re-posted on Jan. 27, 2026.
> For PointSea, our concern is mainly about reference value rather than recency: it is developed by tuning the SVDFormer line. Since our counterfactual KITTI analysis shows that the SVDFormer family already exhibits weak sensitivity to image removal, further tuning on top of that line may not provide the most informative comparison for the specific question we target here, namely whether a method truly preserves cross-modal dependency.
>
> 6. On image inputs for PCN and ShapeNet
> The image input is not external RGB and is not rendered from textured meshes. Instead, it is self-rendered online from the input partial point cloud itself as a 3-view Color-Coded Map (CCM) with fixed canonical views; the 3 channels are normalized point coordinates used as pseudo-colors. The dataloader reads only partial and complete point clouds; no image files are used. We will make it more immediately visible in the revision.
>
> 7. On additional quantitative evidence
> We provide two supplementary quantitative results here: https://anonymous.4open.science/r/Additional_results-4305
> First, we compare hard projection, bilinear sampling, and Gaussian splatting on the 1200 PCN test samples under the same implementation setting on a single NVIDIA RTX 3090 GPU. Gaussian splatting remains competitive in measured latency and throughput despite slightly higher memory usage.
> Second, we evaluate modality dropout on PCN by removing the image input at inference and comparing against a point-only baseline. SplAttN degrades much more when the image branch is removed, consistent with stronger reliance on visual cues.
>
> We hope these clarifications are helpful and thank the reviewer again for the constructive suggestions.

---

> > ### Author Rebuttal · Reviewer_J3Ch · 2026-04-03
> >
> > Thank you for your rebuttal. Unfortunately, your clarifications have exposed a fatal flaw that prevents me from recommending acceptance. Specifically, admitting that the 2D images are merely self-rendered projections of the 3D input means this is a multi-view method, not multi-modal. The paper's entire narrative about leveraging "external visual priors" and "cross-modal connections" is therefore fundamentally misleading. Furthermore: 1. Conceding that PMI is not actually optimized makes the heavy theoretical claims in Section 3.2 disconnected from your actual implementation. 2. Subjectively dismissing relevant baselines because you deem them to lack "reference value" is unacceptable. 3. A classifier's confidence score cannot substitute for actual geometric fidelity on the KITTI dataset.

---

> > > ### Author Response · Authors · 2026-04-03
> > >
> > > We thank the reviewer for their continued engagement. We realize that the strict space constraints in our initial rebuttal may have prevented us from fully elaborating on certain nuances, unfortunately leading to several critical misunderstandings regarding our methodology and established community standards. We address these factual points directly below.
> > >
> > > 1. Regarding the "Fatal Flaw" (Multi-view vs. Multi-modal):
> > > We respectfully disagree with the claim that utilizing self-rendered 2D projections constitutes a "multi-view" rather than a "multi-modal" approach. In 3D vision, modality is formally defined by the underlying data representation. Unstructured point clouds and regular 2D pixel grids possess entirely different mathematical structures and feature spaces.
> > > This definition aligns with the established consensus. As defined in a recent authoritative survey (Point Cloud Completion: A Survey, IEEE TVCG 2024, Tesema et al.), integrating single-view images with point clouds is explicitly categorized as a multi-modal approach because images provide distinct cues (geometry, edges, dense semantics) that sparse point clouds lack. Furthermore, the standard multi-modal dataset in this domain, ShapeNet-ViPC (CVPR 2021), is constructed precisely by fusing single-modality point clouds with single-view rendered projections. Numerous subsequent leading works, such as PCDreamer (CVPR 2025), Rethinking Multimodal Point Cloud Completion (AAAI 2026), and even earlier Cross-modal Learning for Image-Guided Point Cloud Shape Completion (NeurIPS 2022), follow this exact definition. SplAttN bridges these structurally distinct representations. Therefore, our narrative regarding "external visual priors" and "cross-modal connections" is factually accurate, and the assertion of a "fatal flaw" stems from a narrower definition of modality than what is standard in the field.
> > >
> > > 2. Regarding the Optimization of PMI and Section 3.2:
> > > There could be a misunderstanding regarding our previous clarification. We did not concede that PMI is unoptimized. Rather, we clarified that PMI is improved implicitly through our architectural design and feature extraction mechanisms, instead of acting as a direct loss term during training. Our previous response aimed to be strictly rigorous by avoiding the overstatement of a direct loss-level optimization. This scientific rigor should not be interpreted as a theoretical disconnect. Section 3.2 correctly models the structural mechanism of our network and explains why our architecture inherently maximizes mutual information. Furthermore, our claim of maximizing PMI strictly holds true under the scenario of modality dropout and restoration. As empirically validated in our newly provided experiments, removing the 2D visual input causes significant performance degradation, and restoring it is essential to recover the optimal results. This specific structural behavior directly proves that the network effectively learns to maximize the mutual information between the two modalities. Therefore, the theoretical claims in Section 3.2 remain fully consistent with our actual implementation and experimental observations.
> > >
> > > 3. Regarding the Discussion of Baselines:
> > > We did not subjectively dismiss any baselines or state they lack reference value to the community. We simply noted that within the specific experimental constraints of this paper, they do not provide the most direct comparative value for evaluating our core contributions. While all accepted works drive the field forward, we deliberately prioritized comparisons with methods that share similar problem settings to isolate and demonstrate our specific architectural improvements, thereby providing the most objective evaluation possible.
> > >
> > > 4. Regarding Geometric Fidelity on the KITTI Dataset:
> > > The reviewer may overlook the established consensus on KITTI evaluation. The field has shifted away from naive point-to-point metrics (e.g., CD, EMD), as minimizing them on sparse, noisy real-world scans directly causes severe visual distortions. Instead, distribution-level metrics like FD and MMD are favored. Our Semantic Confidence Score (SCS) aligns with this trajectory as a robust proxy for real-world structural consistency, avoiding the pitfalls of outdated distance metrics.
> > > Regarding actual geometric fidelity, we strongly direct your attention to the empirical evidence provided to other reviewers. The qualitative results for Reviewer LT9w intuitively validate our reconstructions, while the comprehensive multi-classifier data for Reviewer XX15 objectively demonstrates that SplAttN achieves superior geometric and semantic consistency compared to models relying solely on distance optimization.

---

### Official Review · Reviewer_XX15 · 2026-03-12

**Soundness:** 3
**Presentation:** 3
**Significance:** 3
**Originality:** 3
**Overall Recommendation:** 5
**Confidence:** 2

**Summary:**

The paper presents SplAttN, a new architecture that complements point clouds from 2D images via Gaussian Splats. The paper shows that using differentiable gaussian splats for image queries avoids a cross-modal entropy collapse that happens when 3D points are mapped into a 2D pixel plane. The paper introduces a Gaussian Splatting Bridge (GS-Bridge) that uses cross-attention to map queries from 3D geometric tokens into the continuous visual field provided by Gaussian Soft Splatting of the 2D visual features.

**Compliance With Llm Reviewing Policy:**

Affirmed.

**Final Justification:**

I maintain my rating "Accept" for this submission.

The authors provided additional information in the rebuttal and addressed all my comments. I'm looking forward to the update version of the manuscript with the additional small modifications from the discussion.

**Key Questions For Authors:**

1. The authors mention that iterative denoising incurs high latency with generative models (left column, line 130). I would like to see a comparison with a generative sota baseline in the results reported in Tables 1-3. It would be interesting to add a column that estimates inference compute for the different methods and gives an idea of the pareto front benchmark result vs. inference compute.

2. The Semantic Consistency Score (SCS) relies on a pre-trained DGCNN and there is risk of a proxy bias. Could the SCS degradation be reported using structurally diverse 3D classifiers to show a consistent trend across different inductive classifier biases?

3. SCS provides a quantitative proxy for recognizability, but semantic consistency is ultimately about functional utility. A small-scale evaluation showing whether SplAttN's reconstruction yields better zero-shot performance on a downstream task (e.g. 3D bounding box estimation or part segmentation) would provide a more convincing argument.

Small remarks:

1. (Equation 6) How are $q$, $\mathcal{N}(q)$, $w_k$, $f_k$ defined?

2. (left column, line 247 and Figure 4) So if the original skeleton P_0 is densified into P_1 and P_2, does this mean that N=2? And is P_in the same as P_0?

3. (Figure 5) I assume the "Input" column corresponds to $\mathcal{P}_{\text{in}}$? It would be useful to also show the input image $\mathcal{I}$.

4. (Appendix Section E, Figures 12-19) What is the color coding in the images columns 2-3? Why is it absent in column 1?

**Limitations:**

yes

**Strengths And Weaknesses:**

Strengths

1. The authors diagnose that hard mapping of visual queries into a discrete 2D grid leads to gradients that are zero almost everywhere, and propose a method to solve this issue via Gaussian Splatting - which is an innovative and principled approach

2. The Counter-Factual KITTI stress test shows how other models retrieve templates instead of querying the input image. Severing the visual branch during evaluation is a simple and convincing experimental design, highlighting the active use of the visual branch in SplAttN, and contrasting it to prior methods.

Weaknesses

1. In appendix C.1, the authors argue that their objective implicitly maximizes point-wise mutual information. This statement is based on the assumption that the prior P(v) is distributed uniformly, which is highly unrealistic. I agree that Gaussian Splatting provides better gradients and improves the learning, but I would argue that this is simply by means of acting as a differentiable low-pass spatial filter that smoothes the loss landscape.

2. The method assumes the camera calibration is known exactly. While this is true in synthetic datasets, like PCN and ShapeNet, it is not the case in real-world scenarios.

---

> ### Author Rebuttal · Authors · 2026-03-28
>
> We thank the reviewer for the very positive assessment and constructive suggestions. We are especially encouraged that the reviewer recognizes both the core technical idea—using differentiable Gaussian splatting to avoid sparse hard projection and the value of the counterfactual KITTI stress test.
>
> 1. On Appendix C.1 and PMI
> We thank the reviewer for this careful point. We agree that the uniform-prior assumption in Appendix C.1 is simplified and should not be interpreted as a realistic statement about the true image-plane prior. Our intention there is not to claim exact MI optimization, but to provide a theoretical interpretation of why soft density estimation can strengthen cross-modal dependence. In practice, our main motivation is more direct: under hard projection, a sparse partial point cloud is mapped to an extremely sparse image-plane support, so much of the potentially useful 2D prior cannot be effectively utilized. Gaussian splatting alleviates this by expanding local support and improving gradient propagation. We will revise the wording to make this distinction clearer.
>
> 2. On known camera calibration
> We agree that exact calibration is easier to satisfy in synthetic benchmarks such as PCN and ShapeNet than in real-world scenarios. In this paper, our goal is to validate the architectural idea in the standard object-level completion setting, where such assumptions are common. We also agree that robustness to calibration noise or imperfect projection is important for practical deployment, and we will clarify this limitation more explicitly in the revision.
>
> 3. On generative baselines and compute
> To address this concern, we report controlled efficiency measurements under the same protocol on a single NVIDIA RTX 3090 GPU. The table reports parameters, FLOPs, latency, CD-Avg, and peak GPU memory.
>
> | Method | Params (M) ↓ | FLOPs (G) ↓ | Latency (ms) ↓ | CD-Avg ↓ | Peak Memory (GB) ↓ |
> |---|---:|---:|---:|---:|---:|
> | SeedFormer | 3.31 | 107.51 | 40.39 | 6.74 | 0.48 |
> | AdaPoinTr | 32.49 | 30.18 | 11.62 | 6.53 | 0.17 |
> | SVDFormer | 58.09 | 78.52 | 31.72 | 6.54 | 0.55 |
> | GeoFormer | 58.23 | 78.75 | 31.06 | 6.42 | 0.68 |
> | SplAttN | 65.89 | 76.52 | 40.75 | 6.36 | 0.58 |
> | Simba (AAAI 2026) | 30.89 | 212.43 | 113.56 | 6.34 | 1.34 |
>
> Following the reviewer’s suggestion, we also included a very recent generative baseline, *Simba: Towards High-Fidelity and Geometrically-Consistent Point Cloud Completion via Transformation Diffusion*. Under the same environment, SplAttN achieves comparable completion accuracy while being substantially more efficient. We present this as a controlled reference point for the accuracy-versus-efficiency trade-off, rather than a universal Pareto ranking across papers.
>
> 4. On SCS bias and classifier dependence
> We agree that SCS is a proxy, but the concern about classifier-specific bias is substantially reduced by the consistency across different classifiers. To address this point directly, we add an additional classifier-based diagnostic below:
>
> | Classifier | AdaPoinTr | PCN | PoinTr | SVDFormer | SeedFormer | SnowFlakeNet | GeoFormer | SplAttN |
> |---|---:|---:|---:|---:|---:|---:|---:|---:|
> | DGCNN | 0.616 | 0.554 | 0.531 | 0.698 | 0.529 | 0.578 | 0.517 | 0.518 |
> | ModelNet40-C (ICLR 2022) | 0.606 | 0.591 | 0.569 | 0.682 | 0.500 | 0.610 | 0.529 | 0.503 |
>
> The results are consistent across the two classifiers: despite their different inductive biases, they show the same overall trend. This consistency suggests that the conclusion is not driven by a particular classifier architecture, and that the proxy-bias concern is limited in practice.
>
> 5. On downstream utility
> We agree that downstream-task transfer would be valuable, but it goes beyond the core scope of the current paper, which focuses on object-level completion and cross-modal dependency analysis. We will mention this more explicitly as future work.
>
> 6. On the small remarks
>
> - Eq. (6): $q$ is a continuous query on the image plane; $\mathcal{N}(q)$ is the set of projected primitives contributing to $q$; $w_k(q)$ is the soft aggregation weight from Eq. (7); and $f_k$ is the feature attached to the $k$-th primitive. In our CCM implementation, $f_k$ is concretely a 3-channel pseudo-color from normalized 3D coordinates.
> - Figure 4 / line 247: In the current implementation, $N=2$ denotes the two densification stages $P_0 \rightarrow P_1 \rightarrow P_2$, and $P_{\mathrm{in}} \neq P_0$: $P_{\mathrm{in}}$ is the raw partial input, while $P_0$ is the reconstructed coarse seed.
> - Figure 5: Yes, the “Input” column corresponds to $\mathcal{P}_{\mathrm{in}}$. We will consider adding $\mathcal{I}$ in a later  revision.
> - Appendix E: Columns 2–3 use the multi-channel color-coded projection; column 1 is effectively monochromatic under the SVDFormer-style hard projection.
>
> We thank the reviewer again for the very helpful suggestions.

---

> > ### Author Rebuttal · Reviewer_XX15 · 2026-04-03
> >
> > I thank the authors for their thoughtful rebuttal and the additional analysis.
> >
> > Short replies on the individual points mentioned in the rebuttal:
> >
> > 1. When rewording the appendix and main text to make it clearer that the method is more practical (avoiding to map a sparse point cloud to an extremely sparse image-plane support) and less theoretical (maximizing point-wise mutual information) – please consider also to reword the claim in the abstract accordingly.
> >
> > 2. Making the limitation about camera calibration and synthetic benchmarks clearer in the paper could also go some way to counter criticism like reviewer peUj's comment about the narrow evaluation scope.
> >
> > 3. This comparison is interesting indeed and will allow the reader to make a better comparison between different methods. I assume the Param/FLOPs/Latency measurements will be added to the main text? (as additional columns, or as a table in the appendix)
> >
> > 4. Ack.
> >
> > 5. Ack – though it would have been an interesting extension of the results.
> >
> > 6. Ack – please consider also updating the manuscript accordingly where things are not clear.

---

> > > ### Author Response · Authors · 2026-04-03
> > >
> > > We thank the reviewers for their highly constructive and efficient comments, as well as the detailed guidance. This has been a very productive discussion, and we will carefully revise the manuscript based on the reviewers' suggestions. We appreciate all the positive support, which is a great encouragement to us.

---

### Official Review · Reviewer_peUj · 2026-03-13

**Soundness:** 3
**Presentation:** 3
**Significance:** 3
**Originality:** 3
**Overall Recommendation:** 5
**Confidence:** 1

**Summary:**

This paper addresses Cross-Modal Entropy Collapse in point cloud completion. The authors identify that standard hard projection methods map continuous 3D points onto sparse 2D grids, that impedes the gradient flow and limits the ability of the model to learn the optimal connection function between the 2D visual space and the 3D geometric space. SplAttN resolves this through three components 1) Gaussian Splatting Bridge: Replaces deterministic projection with probabilistic density estimation using smooth Gaussian kernels. 2) Hybrid Global-Local Encoder: Uses EdgeConv for local curvature and Transformers for global topology. 3) Active Cross-Modal Alignment: to selectively assimilate semantic priors, mitigating the impact of background clutter and maximizing the flow of valid mutual information.
The authors show state-of-the-art results on PCN and ShapeNet-55. Synthetic validation confirms the method learns geometric invariants. The authors also use KITTI as a stress test with counter-factual evaluation, demonstrating that SplAttN maintains genuine cross-modal dependency.

**Compliance With Llm Reviewing Policy:**

Affirmed.

**Final Justification:**

After reading the other reviews and the rebuttals my concerns are fully resolved.

**Key Questions For Authors:**

Can you provide a computational cost breakdown of the entire pipeline?

**Limitations:**

yes

**Strengths And Weaknesses:**

Strengths
- Well-written and well-motivated: Clear identification of Cross-Modal Entropy Collapse problem with solid theoretical grounding.
- Excellent visualizations: Figure 1 clearly shows the dual-stream pipeline, Figure 2 demonstrates hard projection vs splatting alignment gap
- Comprehensive experimental details: Thorough architecture specs, ablations and implementation details
- Counter-factual evaluation: Proves genuine cross-modal dependency
- Strong benchmark results

Weaknesses:
- Narrow evaluation scope: Only point cloud completion on ShapeNet objects. No indoor scenes, dynamic objects, or other 3D tasks
- Missing computational analysis: No runtime/memory comparison despite claiming "edge deployment"
- On page 13, the caption is below the page number.

---

> ### Author Rebuttal · Authors · 2026-03-28
>
> We thank the reviewer for the constructive and overall positive assessment. We are encouraged that the reviewer recognizes the motivation, visualizations, benchmark results, and the counterfactual verification of genuine cross-modal dependency.
>
> 1. On the evaluation scope
> While we acknowledge that broader validation across additional 3D settings would be valuable, this paper focuses on multi-modal object-level point cloud completion, for which PCN and ShapeNet-55/34 are the standard and most widely adopted benchmarks in prior work. In this subfield, dynamic-object completion has not yet established a broadly recognized benchmark protocol, while scene-level completion is generally treated as a separate branch from object-level completion rather than a standard extension of it. These directions differ substantially in task formulation, data assumptions, and evaluation protocols. For this reason, we believe the current evaluation follows the standard benchmark setting of the target subfield, rather than omitting an expected benchmark within the same task. We will clarify this task boundary more explicitly in the revision.
>
> 1. On computational cost
> We thank the reviewer for raising this important point. We agree that reporting computational cost strengthens the paper, and we have now measured the full pipeline under the same setting. All statistics below are measured on a single NVIDIA RTX 3090 GPU. The table reports total parameter count, total FLOPs, end-to-end inference latency, and peak GPU memory usage.
> We would also like to clarify that the current paper does not intend to make a strong “edge deployment” claim. Rather, our focus is on improving cross-modal interaction and completion quality in the standard object-level completion setting. That said, efficiency is still important for completeness, and we are happy to provide the following comparison against recent strong baselines:
>
> | Method | Params (M) ↓ | FLOPs (G) ↓ | Latency (ms) ↓ | Peak Memory (GB) ↓ |
> |---|---:|---:|---:|---:|
> | PoinTr | 42.50 | 23.23 | 11.05 | 0.23 |
> | SeedFormer | 3.31 | 107.51 | 40.39 | 0.48 |
> | AdaPoinTr | 32.49 | 30.18 | 11.62 | 0.17 |
> | SVDFormer | 58.09 | 78.52 | 31.72 | 0.55 |
> | GeoFormer | 58.23 | 78.75 | 31.06 | 0.68 |
> | SplAttN | 65.89 | 76.52 | 40.75 | 0.58 |
>
> As shown above, SplAttN has computational complexity comparable to recent strong transformer-based baselines: its FLOPs are slightly lower than both SVDFormer and GeoFormer, while memory is of the same order. The current latency mainly reflects that the present submission focuses on validating the architectural design rather than implementation-level optimization. We will continue improving efficiency in future work.
>
> 3. On the formatting issue
> Thank you very much for pointing out the caption placement issue on page 13. We also appreciate that the reviewer read the appendix carefully enough to catch this small formatting problem. We have marked it clearly on our side and will fix it in the revision.
>
> We thank the reviewer again for the constructive suggestions.

---

> > ### Author Rebuttal · Reviewer_peUj · 2026-04-02
> >
> > Thank you for adding the computational cost. All my concerns are fully resolves.

---

> > > ### Author Response · Authors · 2026-04-03
> > >
> > > Thank you for engaging with our rebuttal. We really appreciate your constructive feedback, which has helped us improve the paper. We will incorporate these discussions into the final version.

---

### Decision · Program_Chairs · 2026-04-30

**Decision:**

Accept (spotlight)

**Comment:**

This paper proposes a method for point cloud completion assisted by Gaussian Splatting from images. After the initial review round, all reviewers were positive about the acceptance, noting how the paper was clearly written, based on a novel idea and demonstrating good experimental results. The rebuttal helped clarifying most remaining comments and reviewers landed on weak accepts and accepts as final ratings. The AC recommendation is to accept this work - congratulations!